# Collapse Susceptibility Assessment in Taihe Town Based on Convolutional Neural Network and Information Value Method

**Houlu Li [1], Bill X. Hu [1,\*], Bo Lin [2], Sihong Zhu [2], Fanqi Meng [3] and Yufei Li [3]**

1 School of Water Conservancy and Environment, University of Jinan, No.336, West Road of Nan Xinzhuang, Jinan 250022, China; 202121100475@stu.ujn.edu.cn
2 Shandong GEO-Surveying&Mapping Institute in China, Jinan 250013, China
3 Shandong Provincial Territorial Spatial Ecological Restoration Center, Jinan 250000, China
\* Correspondence: stu_huxn@ujn.edu.cn

**Abstract:** The cause mechanism of collapse disasters is complex and there are many influencing factors. Convolutional Neural Network (CNN) has a strong feature extraction ability, which can better simulate the formation of collapse disasters and accurately predict them. Taihe town's collapse threatens roads, buildings, and people. In this paper, road distance, water distance, normalized vegetation index, platform curvature, profile curvature, slope, slope direction, and geological data are used as input variables. This paper generates collapse susceptibility zoning maps based on the information value method (IV) and CNN, respectively. The results show that the accuracy of the susceptibility assessment of the IV method and the CNN method is 85.1% and 87.4%, and the accuracy of the susceptibility assessment based on the CNN method is higher. The research results can provide some reference for the formulation of disaster prevention and control strategies.

**Keywords:** collapse; susceptibility; convolutional neural network (CNN); information value method (IV); Taihe town





## 1. Introduction

Collapse is a geological phenomenon characterized by the detachment and sudden collapse of a steep slope of rock or soil, resulting in the accumulation of material at the foot of the slope or in a valley due to gravity, often leading to significant loss of life and property. Collapses include rock falls, soil falls, and ice falls, which are different from landslides. A collapse occurred on 14 August 2019, at Chengkun Aidai railway station in Suxion, Ganluo County, Sichuan Province, China, where a rock mass detached from a slope, resulting in the death of seventeen workers and blocking railway traffic [1]. On 3 June 1980, a catastrophic collapse occurred at the Yanchihe phosphate mine in Yuan'an County, Hubei Province, claiming the lives of 284 people [2]. Therefore, the susceptibility assessment of collapse disasters can provide support for earthquake prevention, disaster reduction, and engineering construction. With the development of the Geographical Information System (GIS) and Remote Sensing (RS) technology, geological hazard susceptibility assessments have also become more efficient and accurate, which greatly accelerates the research process in this field. The assessment methods of geological hazards mainly include the Analytic Hierarchy Process (AHP) [3–7], the Support Vector Machine (SVM) [8,9], the Information Value method (IV) [10,11], the Frequency Ratio method (FR) [4,12], the Weight of Evidence method (WOE) [4,13,14], Artificial Neural Network (ANN) [15–17], Random Forest (RF) [8,18] and a variety of model coupling methods [19–22].

In recent years, with the significant improvement in the quality and quantity of geological disaster data, deep learning has been widely used in disasters, such as tsunami disasters [23,24], earth prediction [25,26], volcanic eruptions [27,28], et al. But the current hot research is still focused on the field of landslides and land subsidence, rather than collapse. Ding Qing et al. [29] have used Long Short Term Memory (LSTM) to predict

ground subsidence in Wuhan, China. JM et al. [30] have used LSTM, the recurrent neural network (RNN), and the deep neural network (DNN) to make landslide susceptibility maps for Maoxian County, China. Abaker et al. [31] have introduced a deep learning- and IoT-based framework for rock-fall early warning, devoted to reducing rock-fall risk with high accuracy. Tian et al. [32] present a non-contact vision method using deep learning and computer vision technology to study collapse.

As one of the most representative methods in deep learning, a CNN has a strong learning ability to better simulate the formation of disaster and accurately predict the location of potential disaster points. Ge et al. [33] use five kinds of CNN, including AlexNet, Inception-v3, Xception, ResNet-101, and DenseNet-201, to predict landslide susceptibility along a transmission line. Wang et al. [34] have constructed one-dimensional, two-dimensional, and three-dimensional landslide data expression forms, and proposed three CNN models to evaluate landslide susceptibility in Yanshan County, China. Yue et al. [35] used a two-step CNN to classify seismic events. Although there are few studies on the assessment of collapse susceptibility based on convolutional neural networks, this study can explore the applicability of convolutional neural networks in the field of collapse vulnerability evaluation.

In this paper, Taihe Town is taken as the research area. Based on GIS and TensorFlow platforms, multi-source data are used to generate and compare the collapse hazard susceptibility zoning maps based on CNN and IV. On the one hand, this paper provides a basis for the assessment of collapse susceptibility and, on the other hand, explores the applicability of the CNN method in the field of collapse.

The main research schemes are as follows:

(1) Referring to the existing literature and expert experience, we sort previous studies in the study area; collect distance from water system, distance from the road, land cover type, normalized difference vegetation index, planform curvature, profile curvature, slope, aspect, and geological data; and conduct data pre-treatment.

(2) Conduct correlation analysis on the data and eliminate the data with strong correlation. The data are classified, and the information quantity carried by each classification factor is calculated. Based on the information value of each factor, the collapse susceptibility partition map is made by superposition.

(3) Standardize the data and make the data set; Construct the CNN and use data sets for training and validation. The CNN was tested after training. The tested CNN was used to predict the susceptibility of the study area to collapse.

(4) The susceptibility zoning maps based on the two methods were compared using ROC curves.

## 2. Materials and Methods

### 2.1. Overview of the Study Area

The study area, Taihe Town, is located in the southeastern part of Zichuan District, Zibo City, Shandong Province (Figure 1). The area features a crisscrossing network of roads and is an important transportation hub in western and southern Shandong [36]. The town with the largest population and number of villages is located within the coordinates of 118.003° E to 118.252° E, 36.607° N to 36.375° N (Figure 1) [37]. The water system within the Taihe Town area belongs to the Xiaoqing River Basin, which includes two primary rivers, the Zihu and E Zhuang, and 42 secondary rivers such as Dongyuliang, Xiyuju Feng, and Xiangyu, with a river network density of 0.7 km/km$^2$ and a total runoff of 50 million m$^3$. Among them, the Zihu River is the largest river in the area, with a length of 28 km. Taihe Town belongs to a warm-temperate, semi-humid, continental monsoon climate, characterized by four distinct seasons, abundant sunshine, and plenty of rainfall as well as a dry and windy spring, a hot and rainy summer, a cool and dry autumn, and a cold and less snowy winter. The average annual temperature in the area is 12.9 °C, with an average of 2564 h of sunshine per year. The average annual precipitation is 730 mm, with

an average of 52 rainy days per year. The rainfall is concentrated from June to September each year, with the most rainfall occurring in July and August.

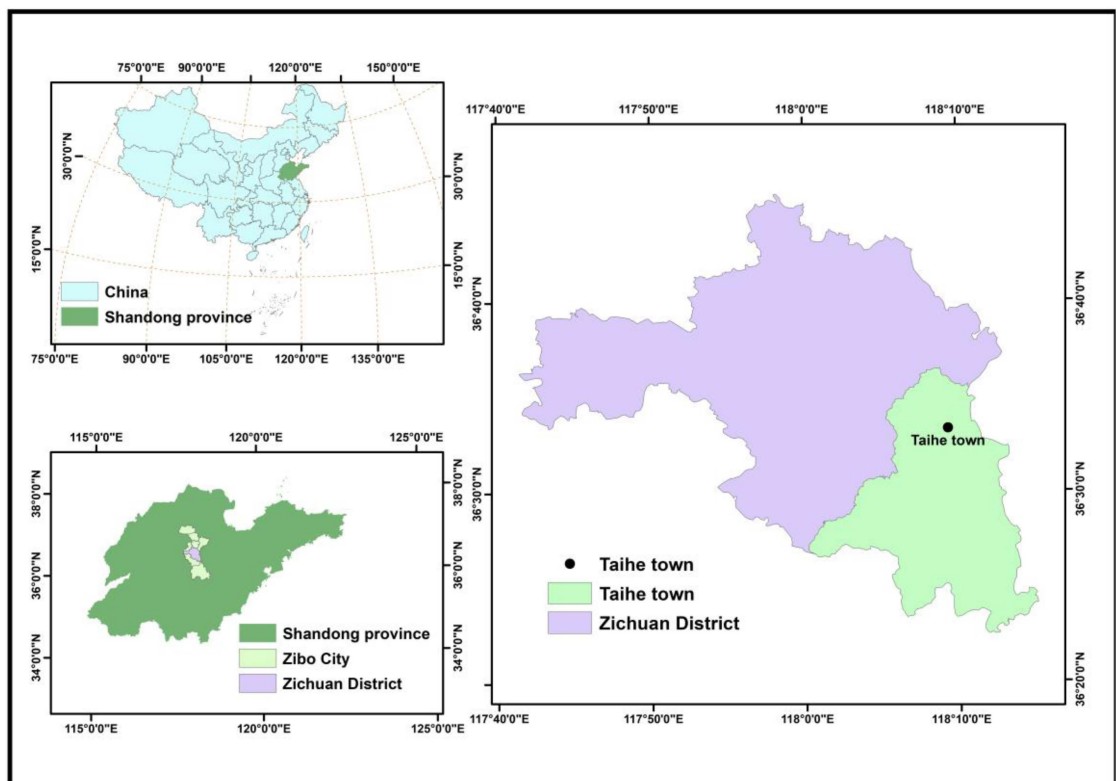

**Figure 1.** Location of study area.

Based on the latest geological hazard field survey results in Zichuan District, there are 53 hazards or disaster points in the Zichuan area, which cumulatively threaten a population of 1767 and property worth 3.705 million RMB (The data came from the project "Study on InSAR Interpretation optical Remote Sensing Interpretation and Early Identification of Geological Hazards in Shandong Province in 2021 Geological Hazard Risk Survey, China" and from the project report "Research on Geological Hazard Identification Based on the combination of Air and Earth Integration and disaster pregnancy Background Indicators (Taking Zichuan District as an example)"). Some pictures of the Taihe town are shown in Figure 2. The collapse sites in Taihe Town are shown in Figure 3.

*2.2. Multi-Source Data*

Cataloged data of geological hazard potential danger points provide information on the location, size, and type of hazards in a particular area and is based on field investigations and compiled from collapse events. The elevation data of 30 m × 30 m resolution used in this analysis is the data source of slope, aspect, planform curvature, and profile curvature. Cataloged data of geological hazard potential danger points and DEM data are provided by Shandong GEO-Surveying&Mapping Institute in China; The 1:250,000-resolution road, river, and geological data are downloaded from the National Catalogue Service For Geographic information (https://www.webmap.cn/, accessed on 25 October 2022); Land cover data is produced from the SinoLC1 dataset and is accessible at https://doi.org/10.5281/zenodo.7707461 (accessed on 4 May 2023). The produced SinoLC1 dataset is the first 1-m resolution and currently the highest resolution land cover product that covers all of China [37]. The Normalized Difference Vegetation Index (NDVI) dataset is provided by the National Ecosystem Science Data Center, National Science & Technology Infrastructure of China. (http://www.nesdc.org.cn, accessed on 6 May 2023). The accuracy and coordinate system of each evaluation factor data are different. Multiple data are unified

into the unit grid with the same resolution of 30 m × 30 m, and the coordinate system is uniformly processed as WGS_1984_UTM_Zone_50N, which is convenient to complete the superposition of multiple factor attribute values.

Susceptibility assessment of collapse hazards requires a good understanding of the environment and trigger factors that induce collapse [38,39]. Representativeness is required for the data used to evaluate the susceptibility of collapse hazards. According to the relevant literature and expert suggestions, this text initially selects nine data variables for research: distance from water system, distance from the road, land cover, NDVI, planform curvature, profile curvature, slope, aspect, and geological data.

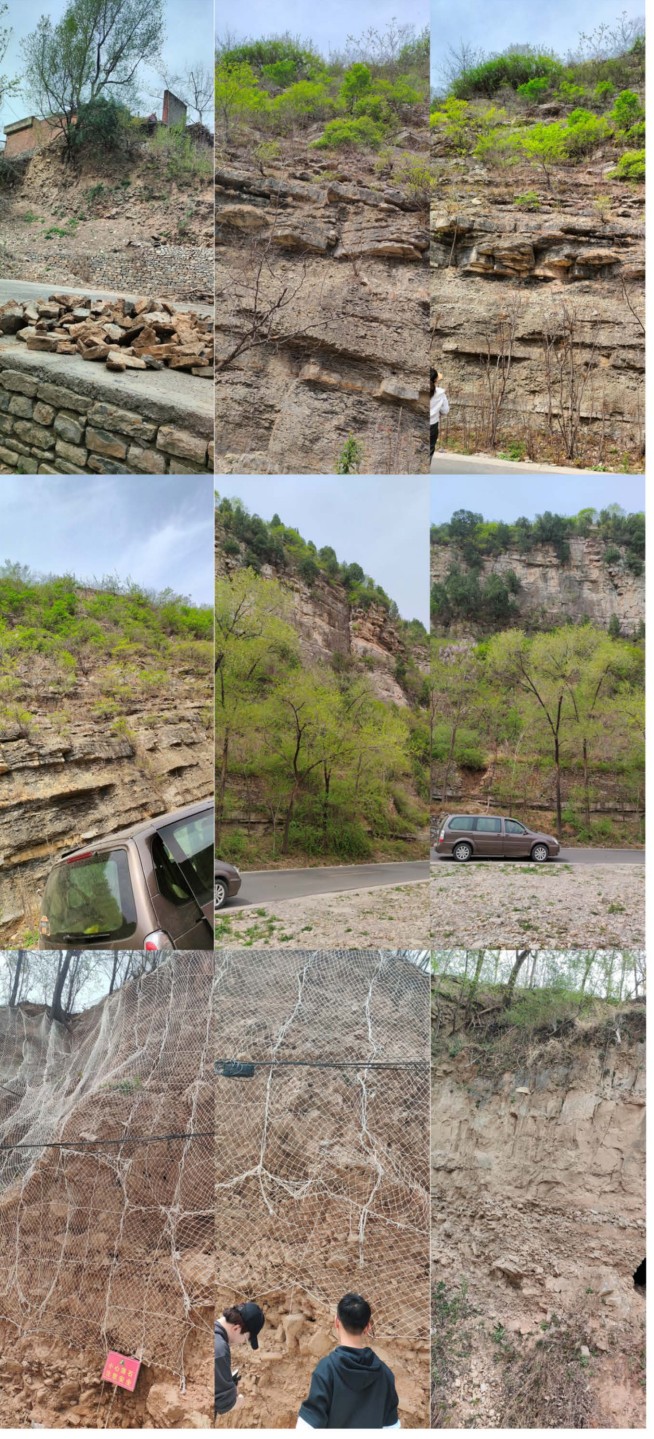

**Figure 2.** Scene of hazard in study area.

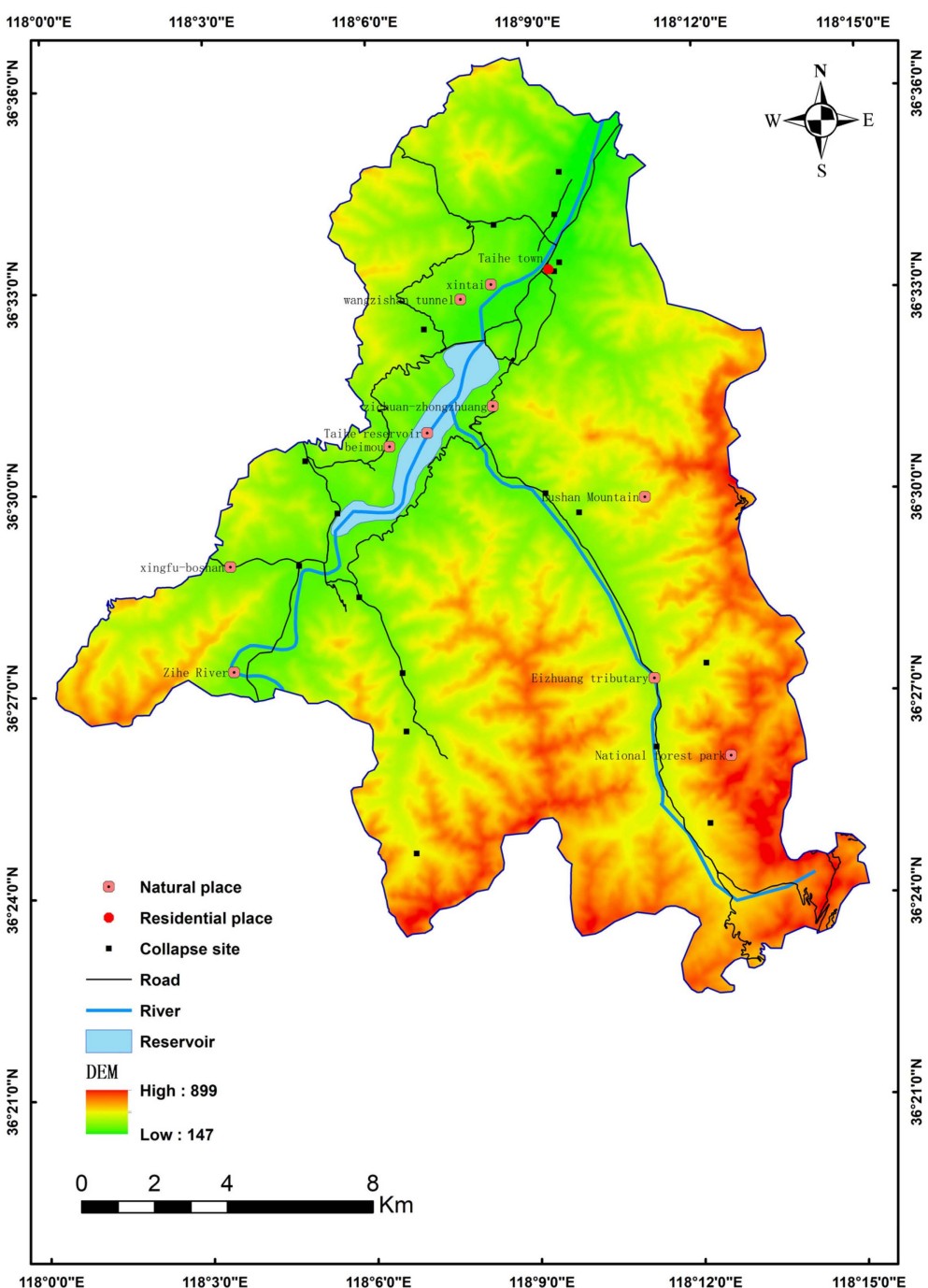

**Figure 3.** The geographical location of the disaster site in Taihe Town.

2.2.1. Distance from Water System

In this study, the water system data we used was the river system. In the assessment of collapse susceptibility, distance from the water system is an important parameter, as the influence of water is one of the main factors leading to collapse. The closer the distance to the water system, the higher the susceptibility to collapse. Specifically, water can cause soil damage through infiltration, gravity, water pressure, and other mechanisms, leading to the occurrence of collapse. Additionally, water can increase the weight of soil, increase soil pressure, and reduce the shear strength of soil, thereby increasing the sensitivity and susceptibility to collapse.

### 2.2.2. Distance from Road

The presence of roads and activities such as cutting slopes for road construction can alter the original stable state of soil and have a significant impact on the stability of the land, thereby affecting the susceptibility to collapse. In general, the closer the distance to roads, the higher the susceptibility to collapse or landslide [40].

### 2.2.3. Land Cover Type

Land cover type is one of the important parameters. Different land cover types have varying effects on the susceptibility to collapse. Specifically, vegetation cover can increase the shear strength and stability of soil and reduce soil erosion and damage, thus reducing the risk of collapse. Therefore, areas with better vegetation cover generally have lower susceptibility to collapse. Conversely, the surface of exposed soil is susceptible to erosion and damage, which can lead to soil looseness and loss, thereby increasing the risk of collapse.

### 2.2.4. Normalized Difference Vegetation Index (NDVI) Data

In the assessment of susceptibility to collapse disasters, the NDVI can be used as one of the indicators for evaluation. The NDVI is an index obtained by calculating the ratio between vegetation infrared reflectance and visible reflectance, which can reflect the vegetation cover on the land surface. Vegetation cover can affect the stability of the land, thereby influencing the susceptibility to collapse. Areas with better vegetation cover generally have lower susceptibility to collapse, while areas with poor vegetation cover are more prone to collapse. Therefore, the NDVI can be used to evaluate the vegetation cover of the land, and thus infer the susceptibility to collapse.

### 2.2.5. Planform Curvature

In the assessment of susceptibility to collapse disasters, planform curvature is one of the important parameters that can be used to evaluate the morphological characteristics and terrain changes of land surfaces, and thus infer the susceptibility to collapse. Planform curvature refers to the curvature radius of the land surface in the horizontal direction and can reflect the changes in the land surface. For land with different planform curvature, the susceptibility to collapse varies. Generally, areas with smaller planform curvature have relatively flat land surfaces and a lower risk of collapse, while areas with larger planform curvature have relatively complex land surfaces and a higher risk of collapse.

### 2.2.6. Profile Curvature

Profile curvature refers to the curvature radius of the land surface in the vertical direction and can reflect the changes in the land surface as well as the potential slope change rate. For land with different profile curvature, the susceptibility to collapse varies. Areas with larger profile curvature have relatively complex land surfaces and a higher risk of collapse.

### 2.2.7. Slope

Slope reflects the degree of inclination and vertical changes of the land surface. For land with different slopes, the susceptibility to collapse varies. Areas with smaller slopes generally have relatively flat land surfaces and a lower risk of collapse, while areas with larger slopes have relatively steep land surfaces and a higher risk of collapse.

### 2.2.8. Aspect

Slope aspect refers to the direction of inclination of the land surface, which can to some extent reflect the direction of natural forces such as water flow and wind on the land surface. For land with different slope aspects, the natural forces acting on it also differ, leading to different levels of susceptibility to collapse. For example, land with a slope aspect facing the direction of rainfall is more susceptible to water erosion and has a higher

risk of collapse. Conversely, land with a slope aspect opposite to the direction of rainfall is less susceptible to water erosion and has a relatively lower risk of collapse.

### 2.2.9. Geological Data

Geological data can directly or indirectly provide information on geological structures, lithology, and other aspects of information, including the composition and structural characteristics of rocks. Different rock types behave differently under stress and deformation. Lithologic data can provide information on the type, distribution, thickness, fractures, and joints of rocks in a certain area, which can help people understand the mechanical properties and stability of rocks, and thus evaluate the likelihood and degree of danger of collapse disasters in that area. Taking the Yihe Formation of Quaternary Holocene in the study area as an example, this rock formation is a fluvial sediment, which is mainly composed of granular materials such as gravel, sand, and mud [41]. The grit particles are relatively large, have high stability, and are relatively not easy to collapse. On the other hand, the shaly sediment has small particles, high water content, and is easy to flow and deform, so it is more likely to collapse when subjected to external forces. The rocks deposited in the Yihe Formation and other fluvial facies often have certain structural characteristics, such as bedding plane, joint, and fissure. These structural features have important effects on the stability and anti-collapse ability of rock mass. If the rock layer tends to tilt, cross, or fracture, it will make the rock mass easy to slide and collapse when subjected to external forces. Joints and fissures may become the occurrence and expansion path of rock mass collapse, which increases the instability of rock mass.

Performing correlation analysis on multiple datasets and eliminating those with high correlation can lead to more accurate evaluation results. This paper uses the Pearson correlation coefficient based on the GIS platform to calculate the degree of correlation between sample features. The Pearson correlation coefficient is generally used to measure the degree of linear correlation between two vectors [42]. Based on the information provided, the dataset "land cover type" has a significant negative correlation with the "NDVI data", with a correlation coefficient of −0.566, which exceeds the threshold of 0.5 (Table 1). In the subsequent processing, "NDVI data" was retained, and "land cover type" data was discarded.

**Table 1.** Multi-source data correlation matrix.

| Layer | 1 | 2 | 3 | 4 | 5 | 6 | 7 | 8 | 9 |
|---|---|---|---|---|---|---|---|---|---|
| 1 | 1.000 | 0.240 | 0.224 | −0.290 | 0.024 | −0.129 | −0.205 | −0.017 | 0.005 |
| 2 | 0.240 | 1.000 | 0.178 | −0.154 | −0.016 | −0.206 | −0.271 | −0.021 | 0.033 |
| 3 | 0.224 | 0.178 | 1.000 | −0.566 | −0.012 | −0.120 | −0.288 | −0.056 | 0.085 |
| 4 | −0.290 | −0.154 | −0.566 | 1.000 | −0.038 | 0.113 | 0.262 | 0.014 | −0.065 |
| 5 | 0.024 | −0.016 | −0.012 | −0.038 | 1.000 | 0.110 | −0.384 | −0.007 | −0.045 |
| 6 | −0.129 | −0.206 | −0.120 | 0.113 | 0.110 | 1.000 | 0.302 | 0.033 | −0.053 |
| 7 | −0.205 | −0.271 | −0.288 | 0.262 | −0.384 | 0.302 | 1.000 | 0.047 | −0.054 |
| 8 | −0.017 | −0.021 | −0.056 | 0.014 | −0.007 | 0.033 | 0.047 | 1.000 | −0.063 |
| 9 | 0.005 | 0.033 | 0.085 | −0.065 | −0.045 | −0.053 | −0.054 | −0.063 | 1.000 |

Notes: Layer: 1. Distance from water system 2. Distance from road 3. Land cover type 4. NDVI data 5. Planform curvature 6. Profile curvature 7. Slope 8. Aspect 9 Geological data.

### 2.3. Methods

#### 2.3.1. Information Value Method

The information value method (IV) is based on statistical models and information theory and calculates the information value of a specific research unit to comprehensively measure the probability of geological disasters. The IV method is a very popular binary statistical method with clear physical meaning, simple operation, and high practicality. It can effectively solve the problem of quantitative evaluation of geological disasters with numerous factors that are difficult to quantify.

The first step is to reclassify each data or influencing factor, and then calculate the information value of each level of influencing factor [43]. In the context of using the IV method for geological hazard susceptibility assessment, reclassifying plays a specific role in simplifying and standardizing the input variables and their associated values. Reclassification involves grouping or categorizing the original data into a smaller number of classes or categories, which facilitates the analysis and interpretation of the variables' contributions to the hazard susceptibility. To avoid subjective bias in the reclassification process, the researcher employed a method of engaging in discussions with other authors. By promoting collaborative discussions and consensus-building among experts, the researcher sought to increase the objectivity of the study. This collaborative process helped reduce the influence of subjective biases, ensuring more reliable and trustworthy results in the reclassification. The total information value is obtained by summing up the information values of multiple factors. The classification of each factor is shown in (Figures 4–11) and its corresponding information value is presented in Table 2. The basic formula is as follows (1), and the total information value formula is as follows (2):

$$I_{ij} = \mathrm{Ln}(\frac{N_i/N}{S_{ij}/S})(i,j = 1, 2 \ldots n) \tag{1}$$

where $I_{ij}$ is the IV of collapse disaster under the class *j* for factor *i*; $N_i$ is the number of disaster points of subcategory *j* under factor *i*; *N* is the total number of hazard points; $S_{ij}$ is the number of grids of subcategory *j* under factor *i*; and *S* is the total number of grids of the study area. The information value of each factor is shown in Table 2.

$$I = \sum_{i=1}^{n} I_{ij} = \sum_{i=1}^{n} \mathrm{Ln}(\frac{N_i/N}{S_{ij}/S}) \tag{2}$$

where, *I* is the total IV of each grid, which is the information value abbreviated. The larger the value of *I*, the more prone the region is to collapse disasters. When using the IV method, each factor has the same weight of 1. The total information content calculated is shown in Figure 12.

**Table 2.** Each data classification and its information value.

| Layers/*i* | Value | *j* | $S_{ij}$ | $N_i$ | $N_i/N$ | $S_{ij}/S$ | *I* |
|---|---|---|---|---|---|---|---|
| | 1 | >300 | 272,871 | 14 | 0.778 | 0.908 | −0.155 |
| | 2 | 0 ≤ 50 | 4638 | 0 | 0.000 | 0.015 | 0.000 |
| Distance from | 3 | 50 ≤ 100 | 4581 | 1 | 0.056 | 0.015 | 1.293 |
| water system | 4 | 100 ≤ 150 | 4610 | 0 | 0.000 | 0.015 | 0.000 |
| | 5 | 150 ≤ 200 | 4630 | 0 | 0.000 | 0.015 | 0.000 |
| | 6 | 200 ≤ 250 | 4561 | 2 | 0.111 | 0.015 | 1.991 |
| | 7 | 250 ≤ 300 | 4587 | 1 | 0.056 | 0.015 | 1.292 |
| | 1 | >300 | 243,257 | 6 | 0.333 | 0.810 | −0.887 |
| | 2 | 0 ≤ 50 | 11,362 | 6 | 0.333 | 0.038 | 2.176 |
| Distance from | 3 | 50 ≤ 100 | 10,125 | 2 | 0.111 | 0.034 | 1.193 |
| road | 4 | 100 ≤ 150 | 9651 | 1 | 0.056 | 0.032 | 0.548 |
| | 5 | 150 ≤ 200 | 9187 | 0 | 0.000 | 0.031 | 0.000 |
| | 6 | 200 ≤ 250 | 8631 | 1 | 0.056 | 0.029 | 0.660 |
| | 7 | 250 ≤ 300 | 8265 | 2 | 0.111 | 0.028 | 1.396 |
| | 1 | −0.2 ≤ 0.012 | 5943 | 0 | 0.000 | 0.020 | 0.000 |
| | 2 | ≤0.228 | 7082 | 1 | 0.056 | 0.024 | 0.857 |
| | 3 | ≤0.313 | 28,487 | 8 | 0.444 | 0.095 | 1.545 |
| NDVI | 4 | ≤0.369 | 79,830 | 4 | 0.222 | 0.266 | −0.179 |
| | 5 | ≤0.421 | 95,414 | 3 | 0.167 | 0.318 | −0.645 |
| | 6 | ≤0.492 | 65,680 | 1 | 0.056 | 0.219 | −1.370 |
| | 7 | ≤0.713 | 18,023 | 1 | 0.056 | 0.060 | −0.077 |

**Table 2.** *Cont.*

| Layers/$i$ | Value | $j$ | $S_{ij}$ | $N_i$ | $N_i/N$ | $S_{ij}/S$ | $I$ |
|---|---|---|---|---|---|---|---|
| Planform curvature | 1 | ≤11.546 | 56,136 | 5 | 0.278 | 0.191 | 0.375 |
| | 2 | ≤20.847 | 64,660 | 5 | 0.278 | 0.220 | 0.234 |
| | 3 | ≤30.790 | 49,622 | 3 | 0.167 | 0.169 | −0.012 |
| | 4 | ≤42.015 | 36,046 | 3 | 0.167 | 0.123 | 0.307 |
| | 5 | ≤54.524 | 26,866 | 0 | 0.000 | 0.091 | 0.000 |
| | 6 | ≤68.636 | 23,884 | 0 | 0.000 | 0.081 | 0.000 |
| | 7 | ≤82.106 | 36,891 | 2 | 0.111 | 0.125 | −0.121 |
| Profile curvature | 1 | ≤2.849 | 78,882 | 10 | 0.556 | 0.268 | 0.728 |
| | 2 | ≤5.249 | 80,357 | 4 | 0.222 | 0.273 | −0.207 |
| | 3 | ≤7.948 | 62,218 | 3 | 0.167 | 0.212 | −0.238 |
| | 4 | ≤10.947 | 38,825 | 1 | 0.056 | 0.132 | −0.865 |
| | 5 | ≤14.546 | 21,662 | 0 | 0.000 | 0.074 | 0.000 |
| | 6 | ≤19.795 | 9640 | 0 | 0.000 | 0.033 | 0.000 |
| | 7 | ≤38.390 | 2521 | 0 | 0.000 | 0.009 | 0.000 |
| Slope | 1 | ≤6.516 | 50,050 | 7 | 0.389 | 0.168 | 0.837 |
| | 2 | ≤11.869 | 63,884 | 7 | 0.389 | 0.215 | 0.593 |
| | 3 | ≤16.989 | 62,680 | 3 | 0.167 | 0.211 | −0.235 |
| | 4 | ≤21.877 | 53,944 | 0 | 0.000 | 0.181 | 0.000 |
| | 5 | ≤27.229 | 39,417 | 0 | 0.000 | 0.133 | 0.000 |
| | 6 | ≤34.211 | 21,192 | 1 | 0.056 | 0.071 | −0.249 |
| | 7 | ≤59.346 | 6093 | 0 | 0.000 | 0.020 | 0.000 |
| Aspect | 1 | flat | 1864 | 0 | 0.000 | 0.006 | 0.000 |
| | 2 | north | 39,613 | 2 | 0.111 | 0.133 | −0.182 |
| | 3 | northeast | 38,874 | 1 | 0.056 | 0.131 | −0.856 |
| | 4 | east | 34,563 | 6 | 0.333 | 0.116 | 1.053 |
| | 5 | southeast | 34,731 | 2 | 0.111 | 0.117 | −0.050 |
| | 6 | South | 32,863 | 0 | 0.000 | 0.111 | 0.000 |
| | 7 | southwest | 33,422 | 5 | 0.278 | 0.112 | 0.904 |
| | 8 | west | 37,812 | 0 | 0.000 | 0.127 | 0.000 |
| | 9 | northwest | 43,518 | 2 | 0.111 | 0.146 | −0.276 |
| Geological data | 1 | νδfK↓1↑1 | 13,415 | 0 | 0.000 | 0.045 | 0.000 |
| | 2 | χCK↓1↑4 | 468 | 0 | 0.000 | 0.002 | 0.000 |
| | 3 | ∈↓4→O↓1→J∠s | 54,927 | 2 | 0.111 | 0.183 | −0.498 |
| | 4 | s | 4788 | 0 | 0.000 | 0.016 | 0.000 |
| | 5 | νδfK↓1↑1 | 1406 | 0 | 0.000 | 0.005 | 0.000 |
| | 6 | ∈↓3→J∠zˆ | 10,006 | 0 | 0.000 | 0.033 | 0.000 |
| | 7 | ∈↓2→Cˆ∠zˆ | 914 | 0 | 0.000 | 0.003 | 0.000 |
| | 8 | ∈↓2–3→Cˆ∠m | 4341 | 0 | 0.000 | 0.014 | 0.000 |
| | 9 | ∈↓3–4→J∠g | 24,371 | 4 | 0.222 | 0.081 | 1.008 |
| | 10 | ∈↓4→O↓1→J∠cˆ | 80,778 | 3 | 0.167 | 0.269 | −0.478 |
| | 11 | Qh∠y | 25,527 | 8 | 0.444 | 0.085 | 1.655 |
| | 12 | O↓2→Mw | 12,826 | 0 | 0.000 | 0.043 | 0.000 |
| | 13 | O↓2→Mt | 3189 | 0 | 0.000 | 0.011 | 0.000 |
| | 14 | O↓2→Mt-w | 801 | 0 | 0.000 | 0.003 | 0.000 |
| | 15 | O↓2→Md | 9367 | 0 | 0.000 | 0.031 | 0.000 |
| | 16 | O↓2→Md-b | 19,719 | 0 | 0.000 | 0.066 | 0.000 |
| | 17 | O↓2→Mb | 12,154 | 0 | 0.000 | 0.040 | 0.000 |
| | 18 | O↓2→Mb-w | 1315 | 0 | 0.000 | 0.004 | 0.000 |
| | 19 | O↓1→J∠s⊥↑a-b | 20,166 | 1 | 0.056 | 0.067 | −0.189 |

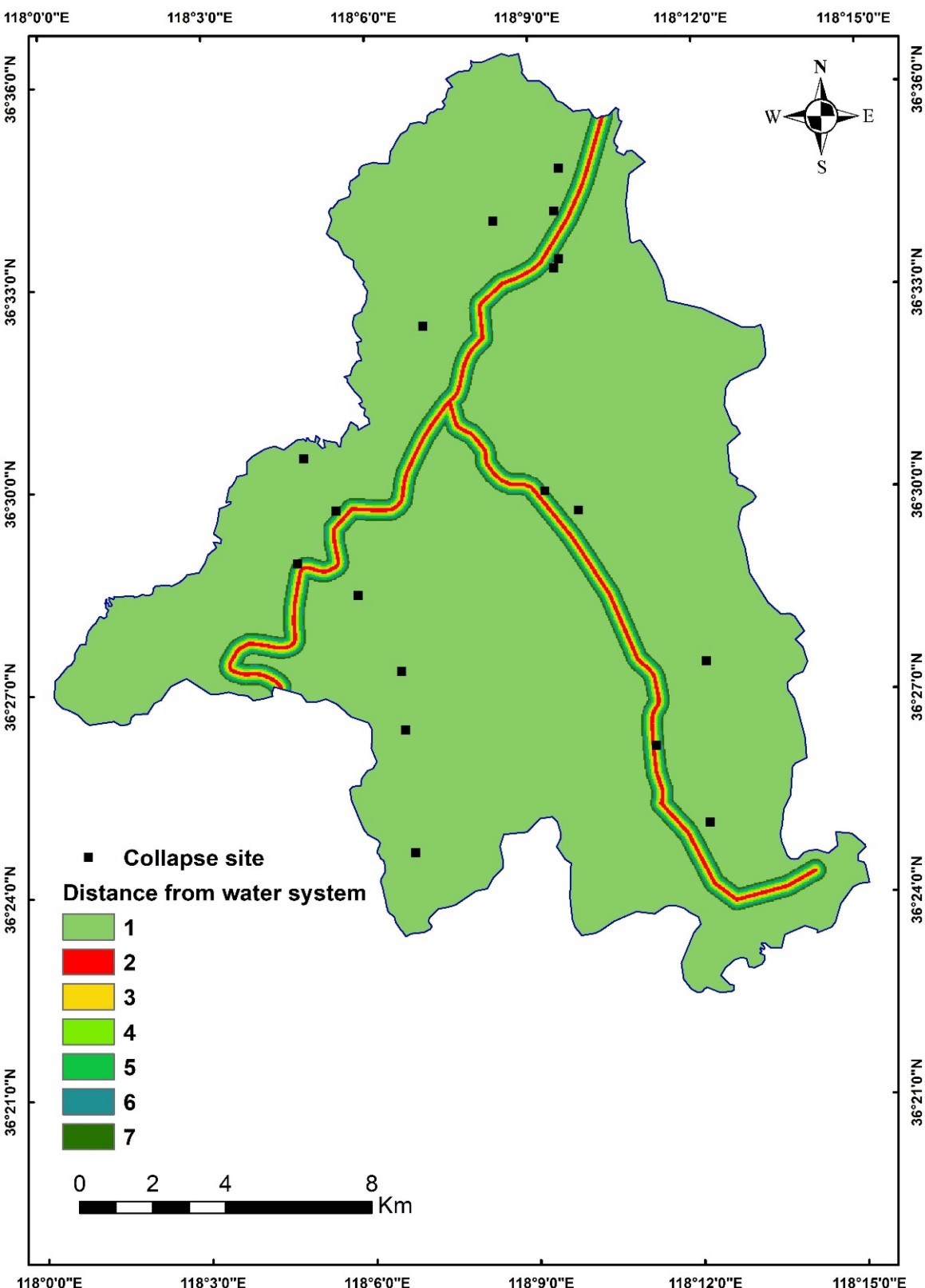

**Figure 4.** Distance from water system.

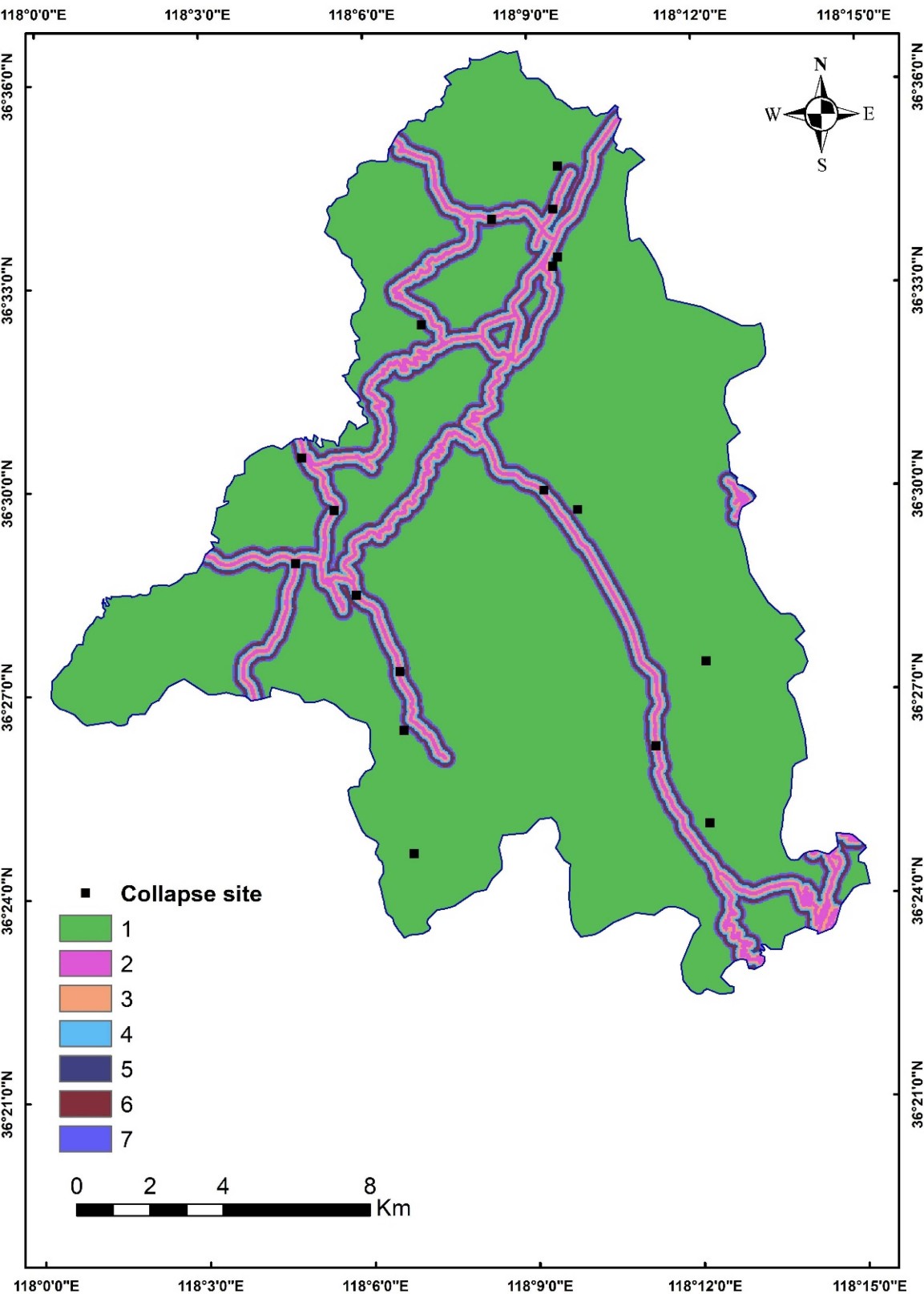

**Figure 5.** Distance from road.

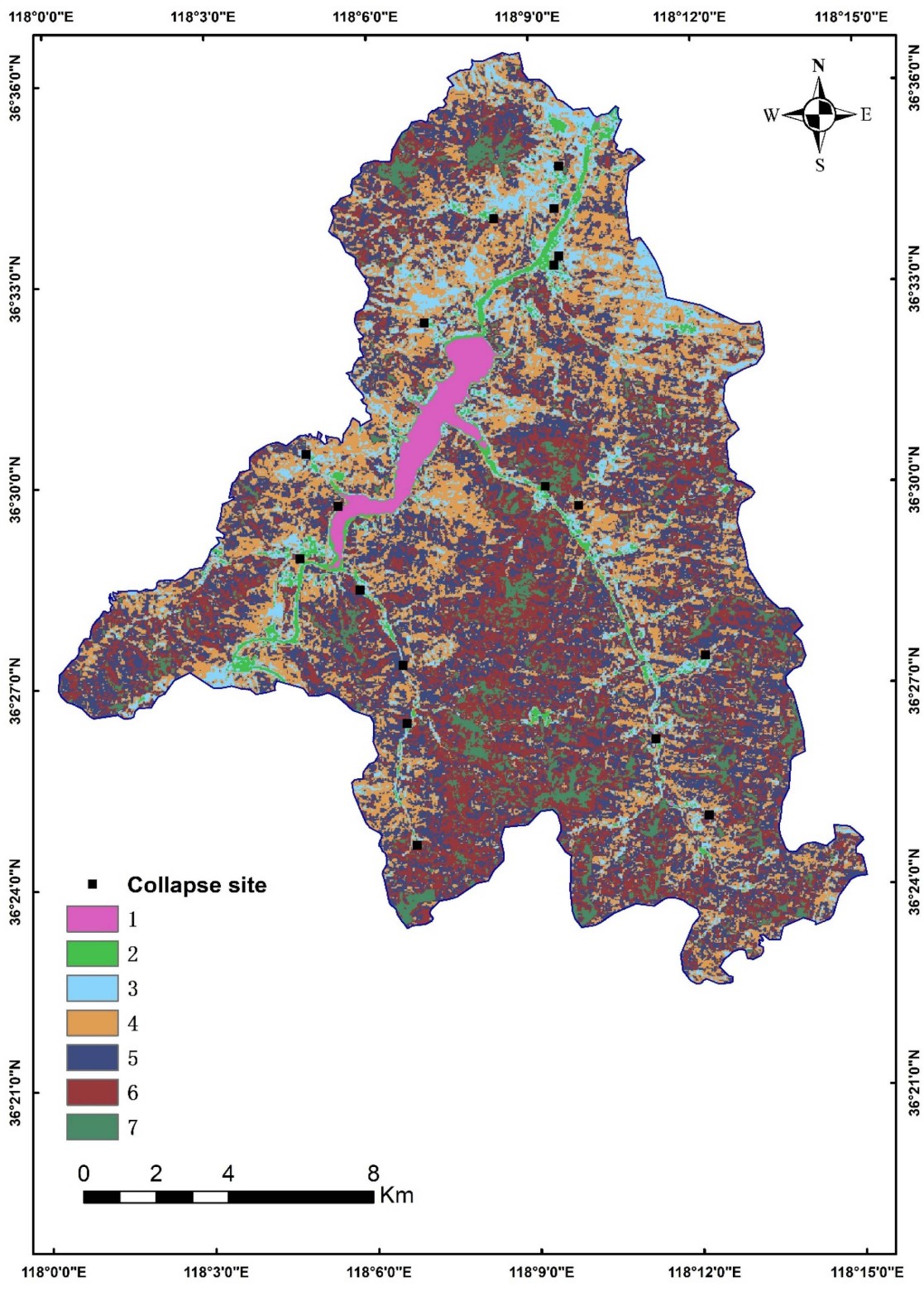

**Figure 6.** NDVI.

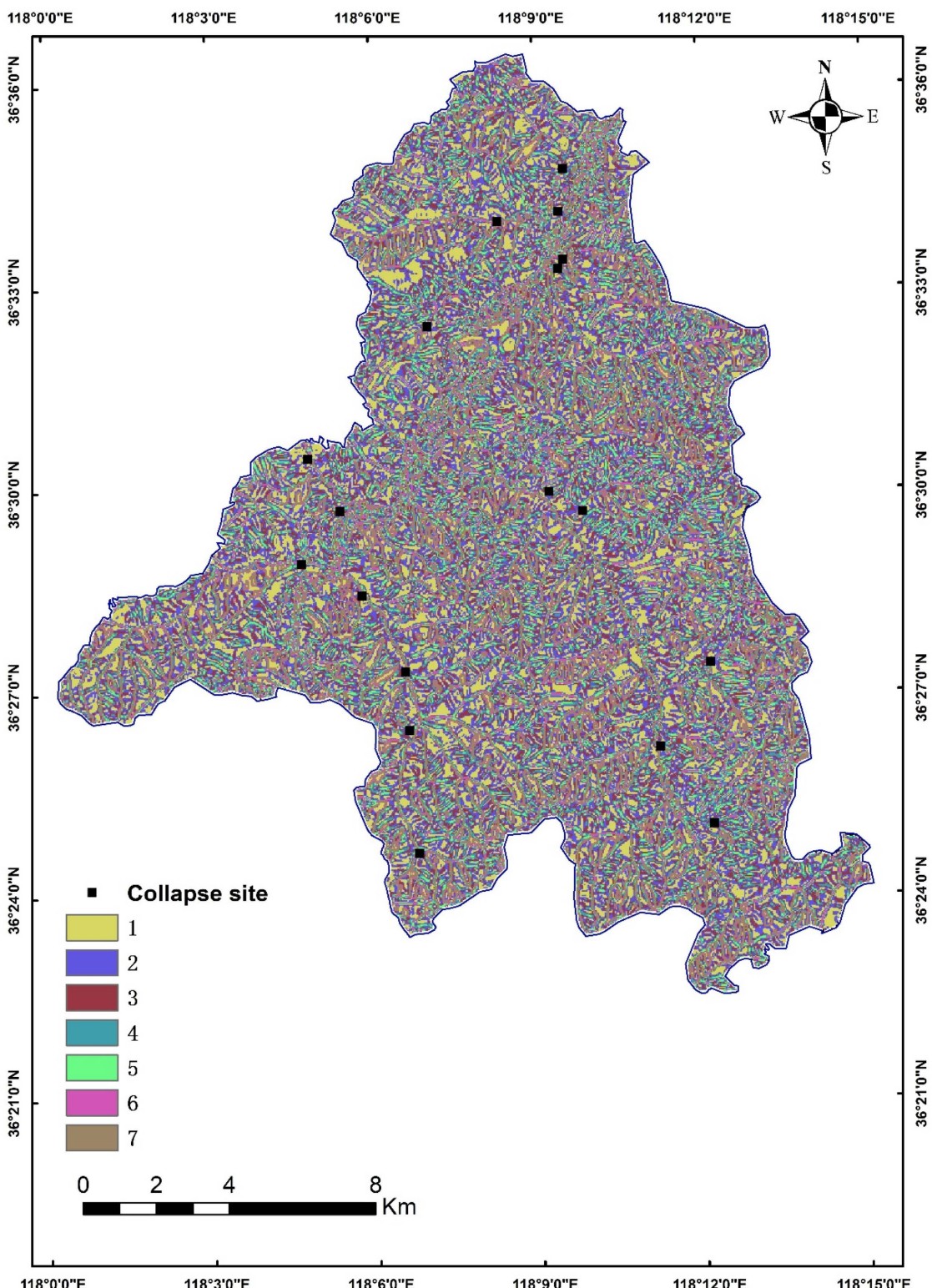

**Figure 7.** Planform curvature.

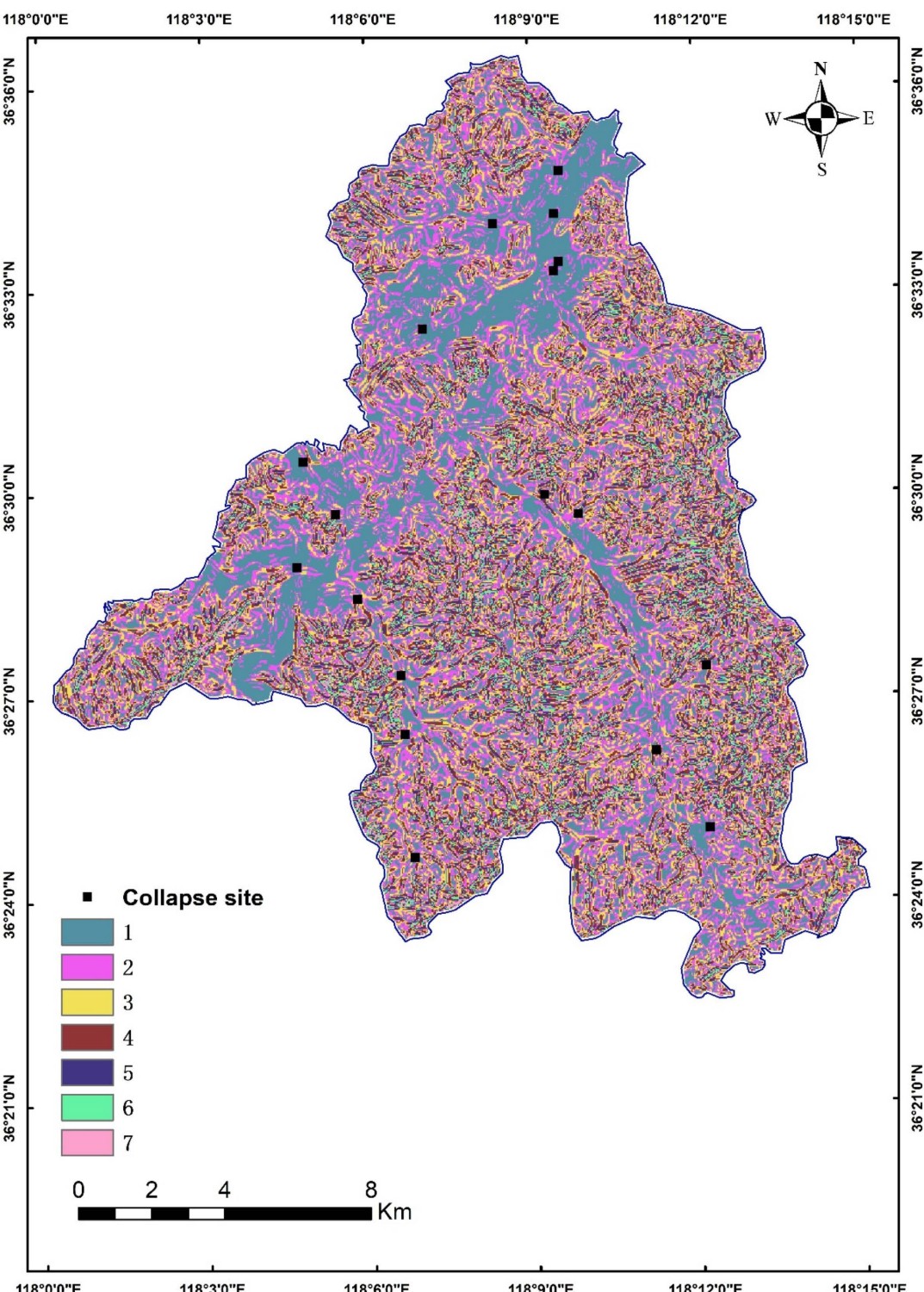

**Figure 8.** Profile curvature.

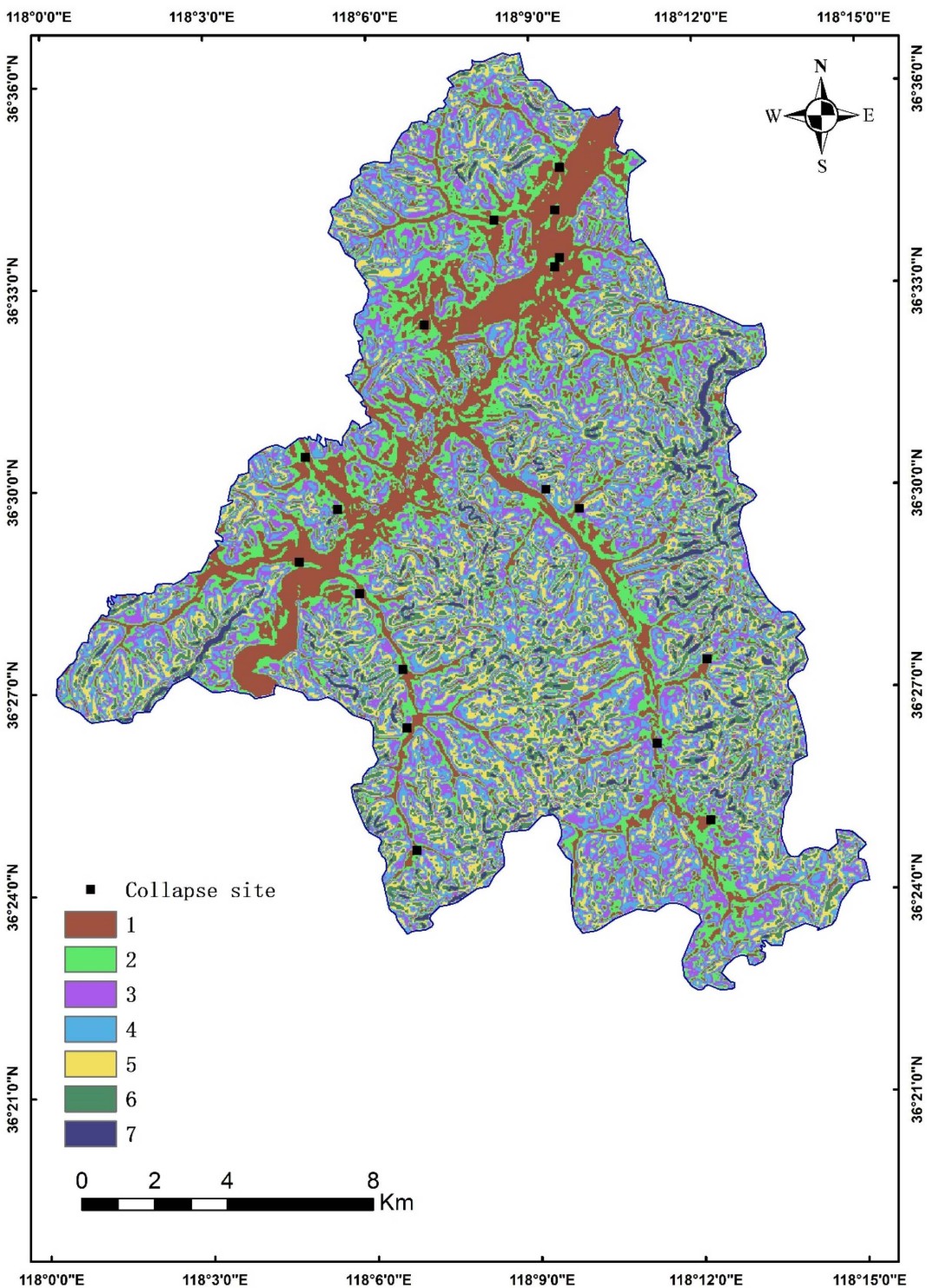

**Figure 9.** Slope.

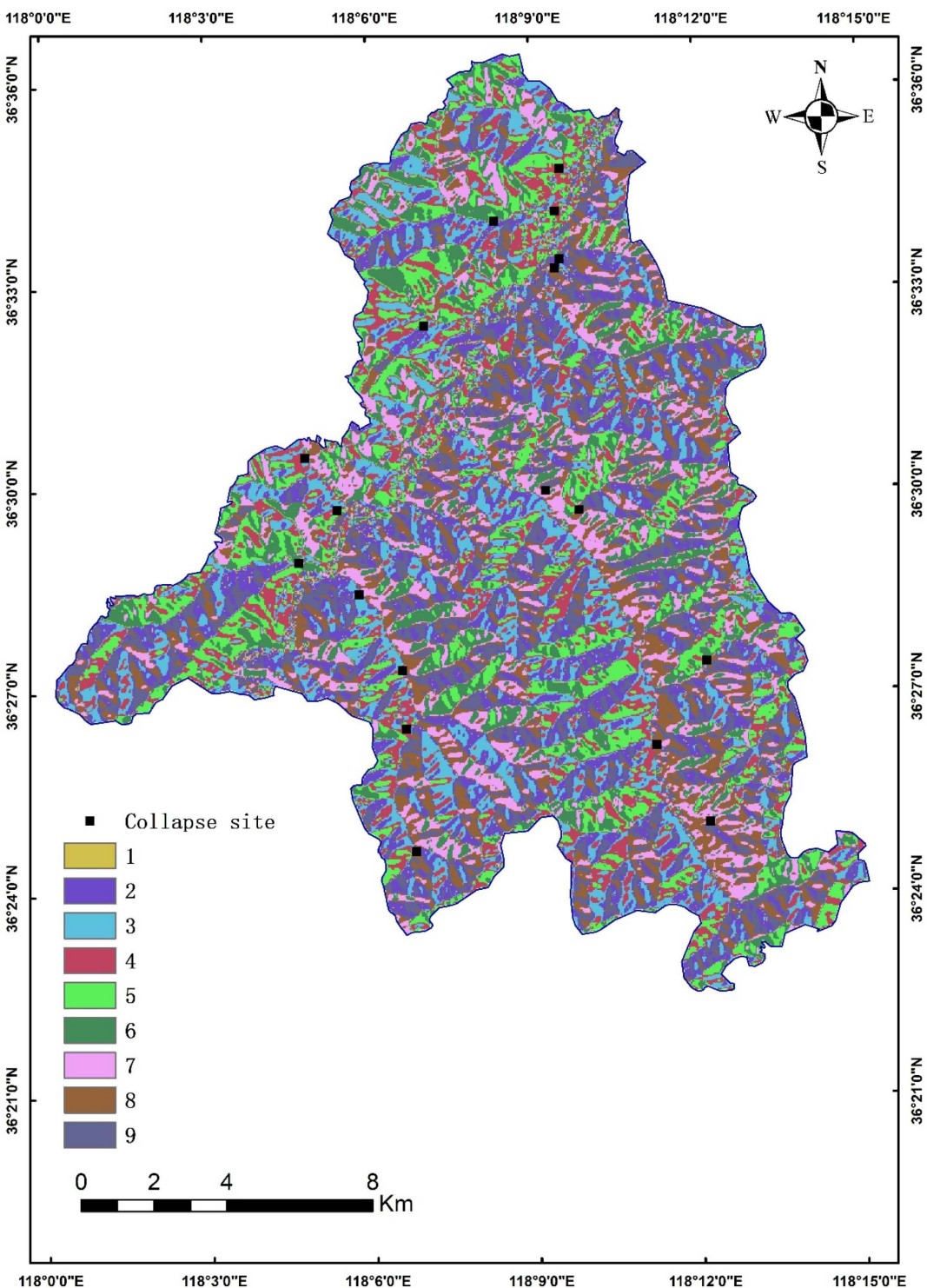

**Figure 10.** Aspect.

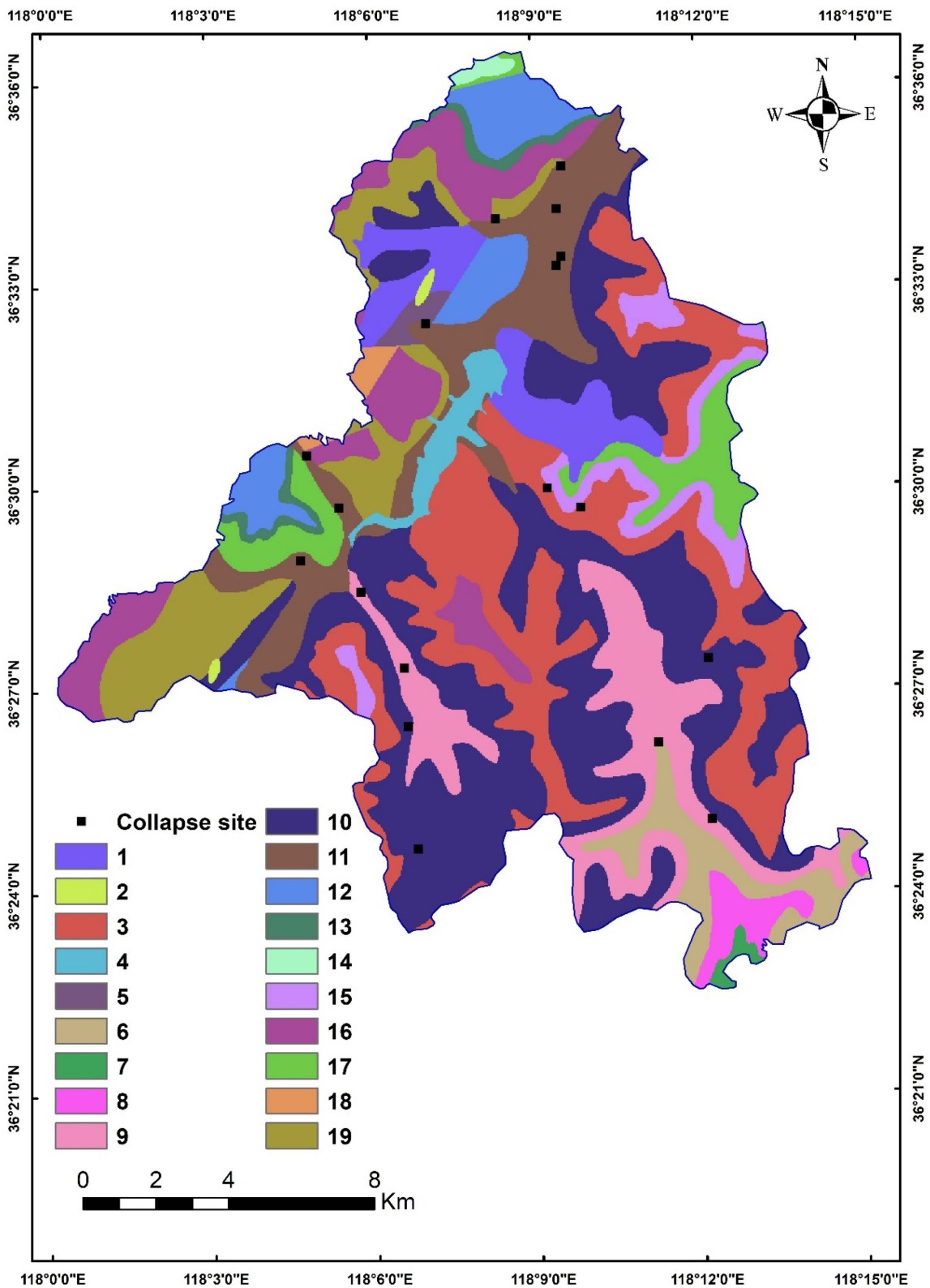

**Figure 11.** Geological data.

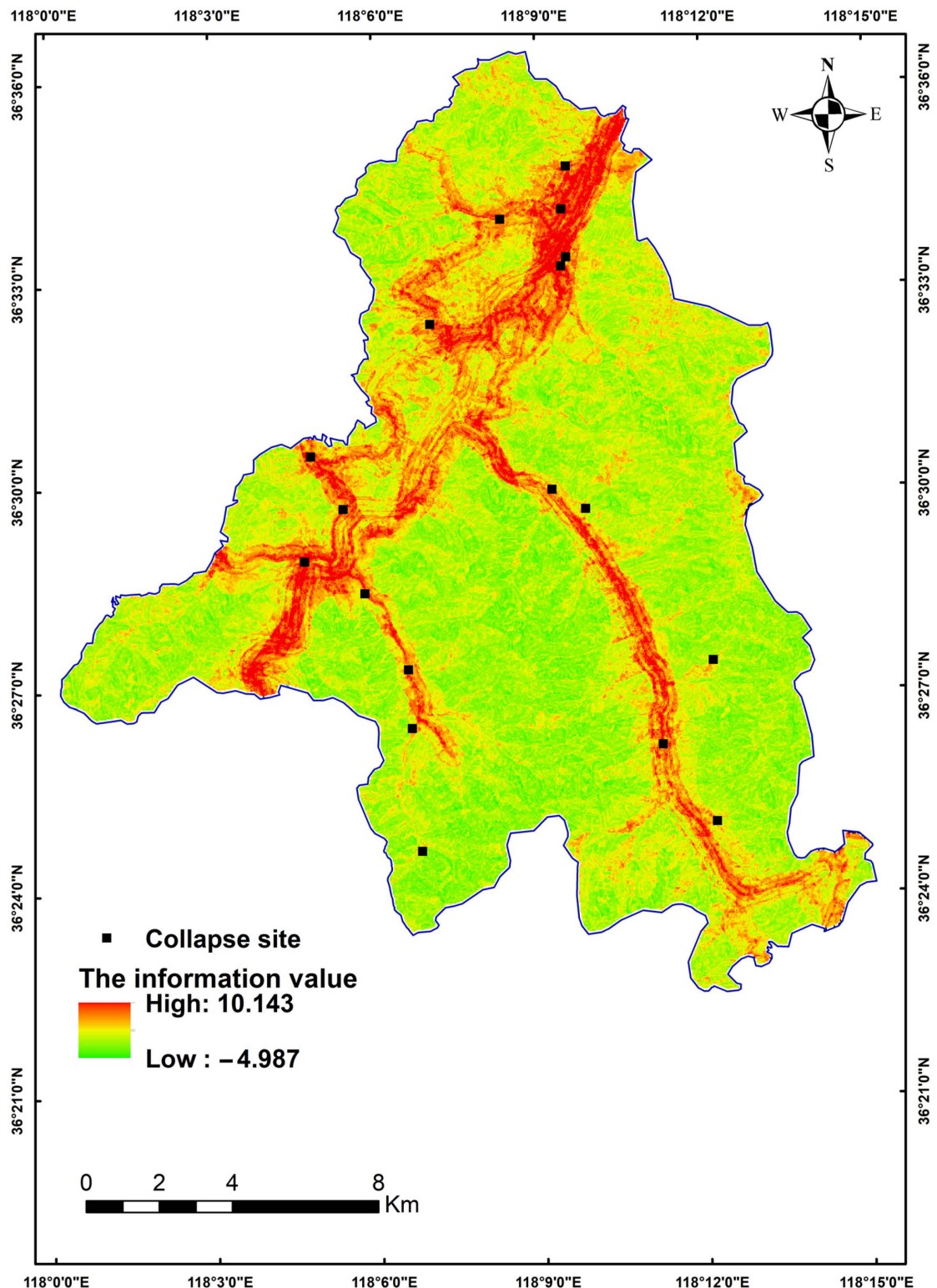

**Figure 12.** Collapse susceptibility zoning map based on IV.

### 2.3.2. Convolutional Neural Network

The CNN consists of a convolutional layer, a pooling layer, and a fully connected layer. These additional layers enable the CNN to effectively process and analyze high-dimensional data by extracting relevant features at different scales and levels of abstraction.

The convolutional layer in the CNN uses a convolution kernel to extract features from the input data. By convolving the kernel over the input data, the network can identify and capture important features of the data, such as edges, shapes, and textures. The basic calculation formula is as follows (3):

$$C_j = \sum_i^N f(\omega_j * x_i + b_j) \ j = 1, 2, 3, 4, \ldots k \tag{3}$$

In the equation, $k$ represents the number of convolution kernels; $C_j$ represents the output of the $j$-th convolution kernel; f represents the non-linear activation function; i represents the spatial position of the convolution operation; $x_i$ represents the input data corresponding to the convolution window; and $\omega_j$ and $b_j$ represent the weight and bias, respectively.

The pooling operation plays a crucial role in reducing the dimensionality of the output feature maps generated by the convolutional layer. By downsampling the feature maps, pooling helps to reduce the number of parameters between layers, thereby reducing model complexity and improving computational efficiency. The function prototype is as follows (4):

$$L_{out} = \left[ \frac{L_{in} + 2 * padding - dilation * (kernel_{size} - 1) - 1}{stride} + 1 \right] \tag{4}$$

In the equation, if the input data shape is (N, C, $L_{in}$), and the output shape is (N, C, $L_{out}$), the $kernel_{size}$ is the size of the sliding windows; stride—the *stride* of the sliding window must be >0; *padding*—implicit negative infinity padding to be added on both sides; *dilation*—the stride between elements within a sliding window.

### 2.3.3. Data Set Construction

Neural networks tend to perform better with standardized and larger datasets. To standardize the multi-source data, the values of each layer are normalized to the range of 0–1. Due to the lack of convenient data for precise disaster points, the available disaster point locations were used to expand the dataset. The collapse point of Taihe town is mainly a small collapse, and the specific range of each collapse point is not all in a 30 m × 30 m grid. According to the first law of geography, "all things are related to other things, but things near are more related than things far away". By considering points that are close to the collapse point and have similar environmental conditions or characteristics, the introduction of irrelevant information and noise can be effectively reduced. Therefore, we choose the eight selected points within the grid selected around the hazard point coordinates as a supplement. Specifically, 8 points were selected around each of the original 18 collapse hazard points, resulting in a total of 162 collapse hazard points. Additionally, 162 non-collapse hazard points were randomly selected outside of a 1 km buffer zone. In total, there were 324 points. The locations are shown in Figure 13.

Based on the expanded collapse point data, a one-dimensional representation was constructed for training and validating the convolutional neural network. The researchers used 70% of the data for model training and 30% of the data for model validation. The data representation is shown in Figure 14.

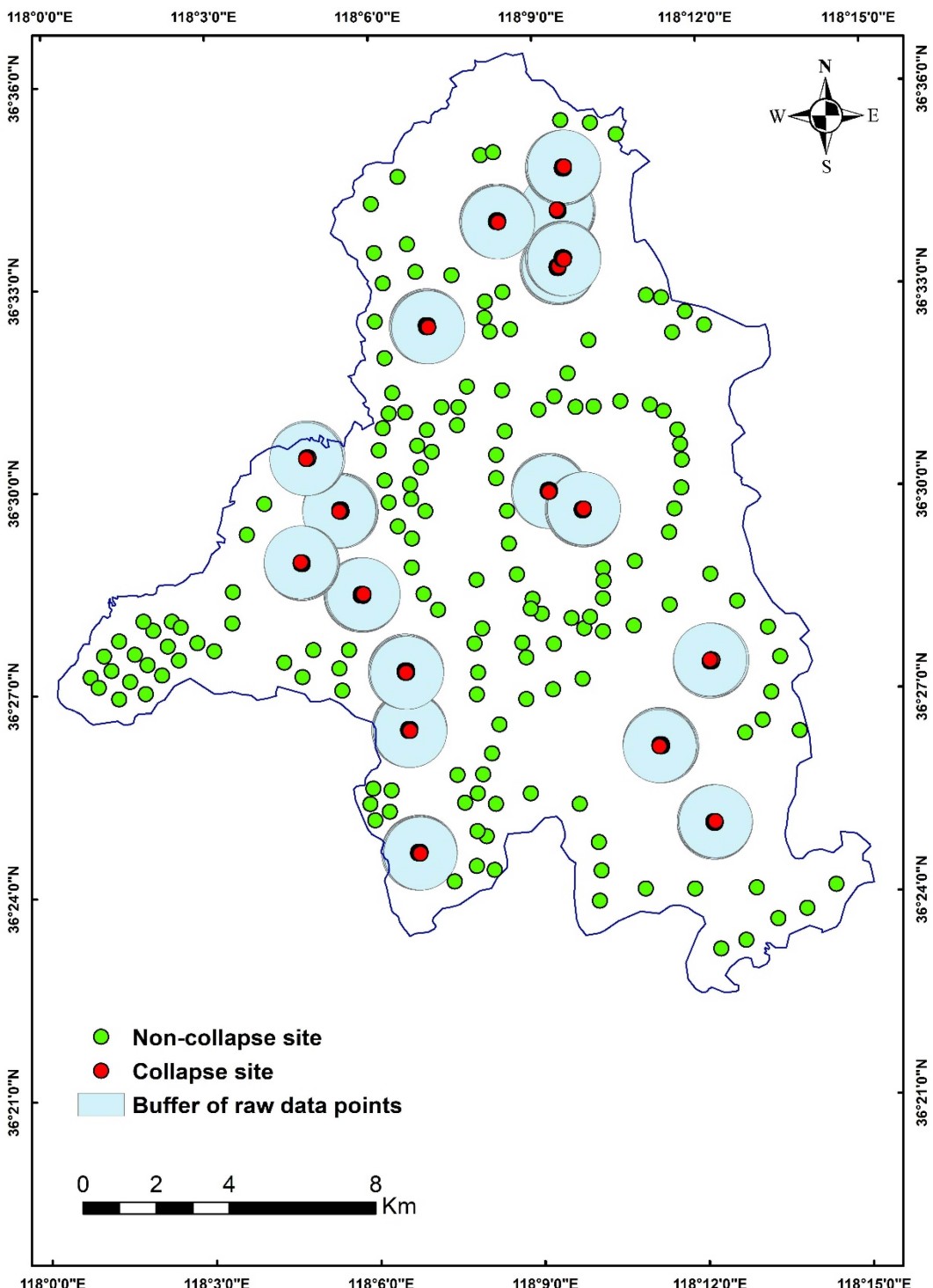

**Figure 13.** Expanded collapse sites.

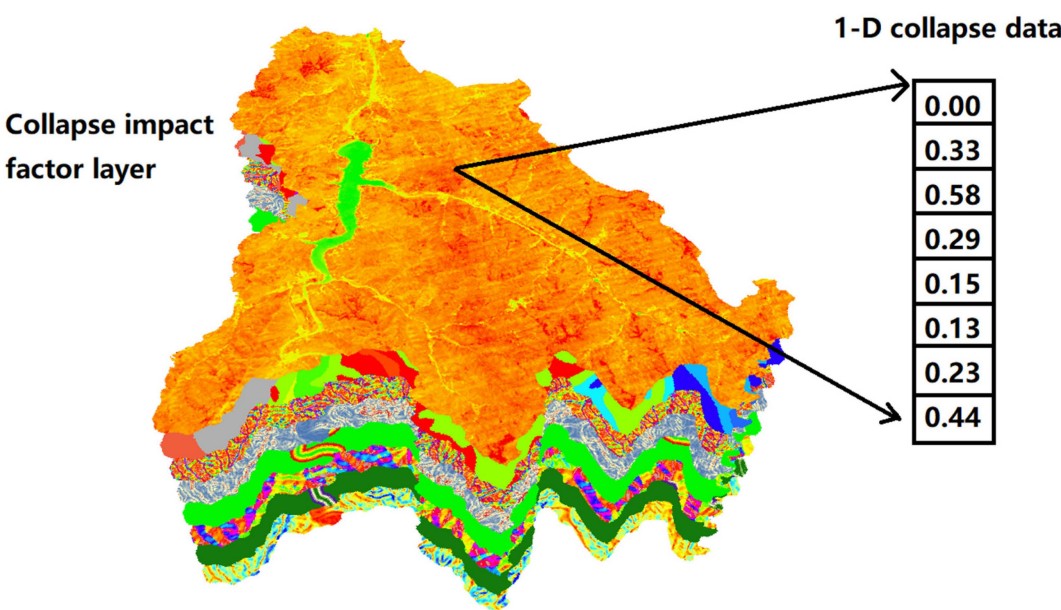

**Figure 14.** One-dimensional disaster data.

### 2.3.4. CNN Construction

In this study, the CNN was built based on the tensorflow2.5.0 machine learning platform. The parameters of each layer are shown in Table 3, with a total of 14,657 parameters, each of which was involved in the training process. The first layer is a convolutional layer, which is also the input layer, with 32 neurons and 32 bias parameters, using a $3 \times 1$ kernel size. The second layer is also a convolutional layer, with 64 neurons and 64 bias parameters, followed by a 1D max pooling layer. The third layer is a Flatten layer, which converts the input data from multiple dimensions to a one-dimensional format by flattening the data. The fourth layer is a dense layer with 64 neurons.

**Table 3.** Convolutional neural network structure and parameters.

| Layer (Type) | Output Shape | Param # |
|:---:|:---:|:---:|
| conv1d (Conv1d) | (None, 6, 32) | 128 |
| conv1d_1 (Conv1d) | (None, 6, 32) | 6208 |
| max_pooling1d (MaxPooling1D) | (None, 2, 64) | 0 |
| flatten (Flatten) | (None, 128) | 0 |
| dense (Dense) | (None, 128) | 8256 |
| dropout (Dropout) | (None, 128) | 0 |
| dense_1 (Dense) | (None, 128) | 65 |
| Total params: 14,657 | | |
| Trainable params: 14,657 | | |
| Non-trainable params: 0 | | |

### 2.3.5. The CNN Training and Verification

The loss curves for the training and validation sets are shown in Figure 15. The model parameters were saved at the 23rd epoch, where the minimum loss value of 0.3392 was achieved and the validation accuracy was 0.8367. The training process was halted when the validation loss had increased for five consecutive epochs, and the optimal parameters were saved.

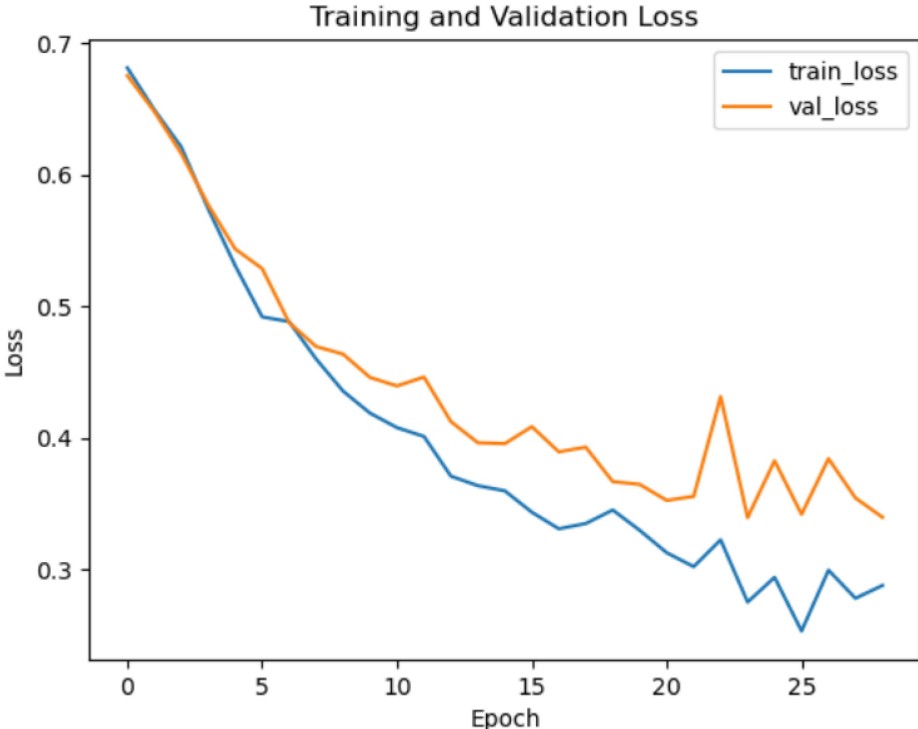

**Figure 15.** Training and Validation Loss.

The confusion matrix is made on the training set and the verification set (Figures 16 and 17), and the abbreviations are shown in Table 4:

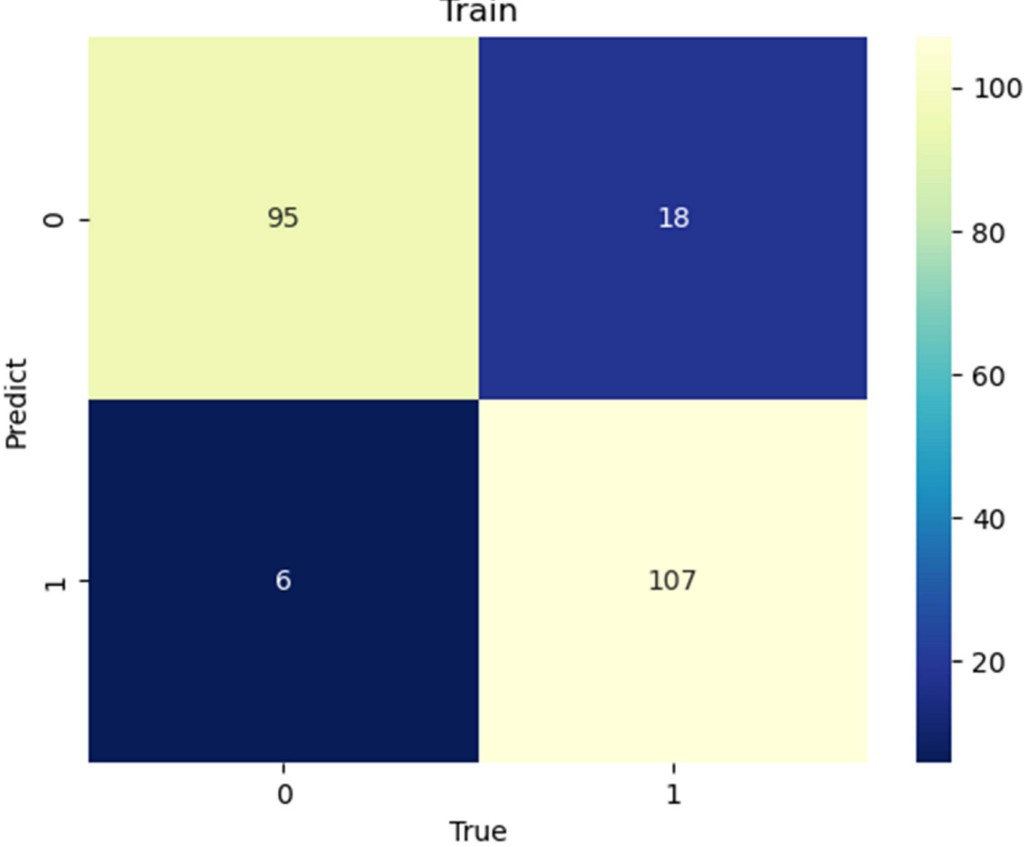

**Figure 16.** Confusion matrix of train dataset.

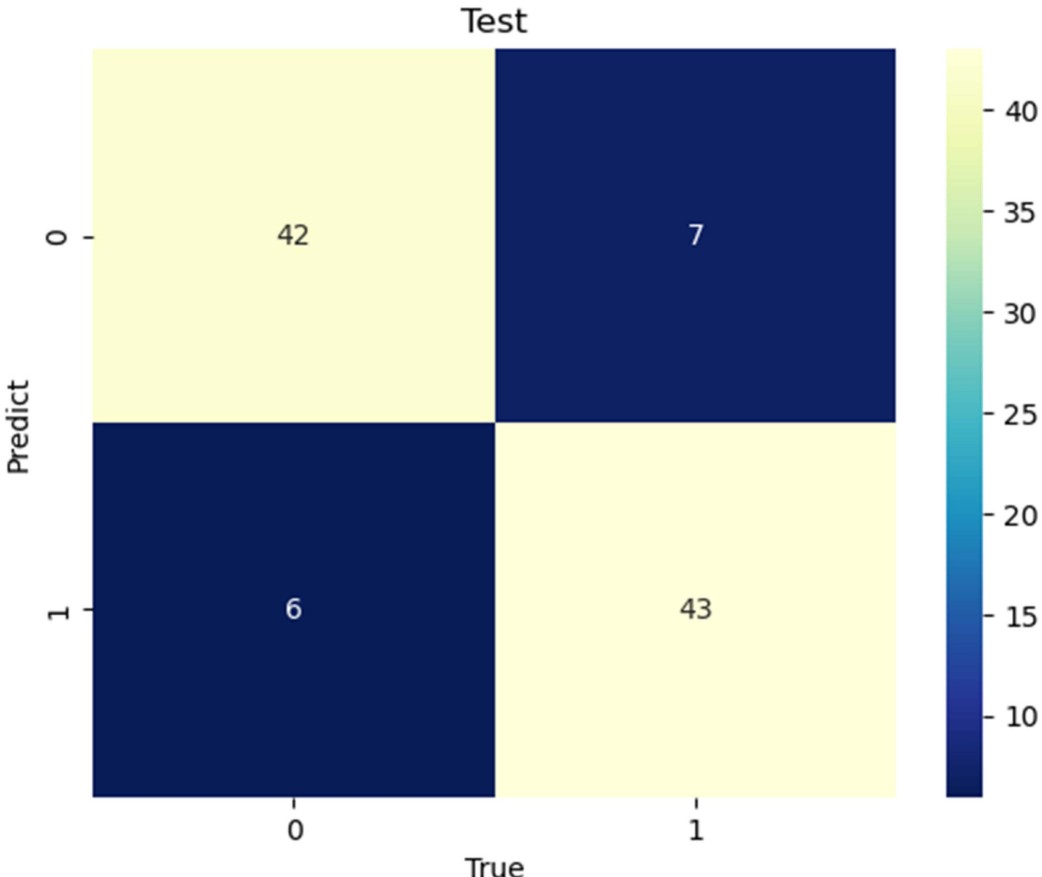

**Figure 17.** Confusion matrix of validation dataset.

**Table 4.** Confusion matrix.

| Confusion | | Predicted Value | |
|---|---|---|---|
| | | **Positive Example** | **Negative Example** |
| True value | Positive example | TP | FN |
| | Negative example | FP | TN |

True Positive (TP): The number of positive cases that were correctly predicted by the model. This refers to the situation where the true label of the data is positive, and the predicted label is also positive.

True Negative (TN): The number of negative cases that were correctly predicted by the model. This refers to the situation where the true label of the data is negative, and the predicted label is also negative.

False Positive (FP): The number of negative cases that were incorrectly predicted as positive by the model. This refers to the situation where the true label of the data is negative, but the predicted label is positive.

False Negative (FN): The number of positive cases that were incorrectly predicted as negative by the model. This refers to the situation where the true label of the data is positive, but the predicted label is negative.

To further evaluate the confusion matrix, the precision, recall, and F1-score metrics were used.

*Precision*: Precision, also known as positive predictive value, measures the proportion of predicted positive cases that are actually positive. It is calculated as (5):

$$Precision = \frac{TP}{TP + FP} \tag{5}$$

*Recall* : Recall, also known as sensitivity or true positive rate, measures the proportion of actual positive cases that are correctly identified as positive by the model. It is calculated as (6):

$$Recall = \frac{TP}{TP + FN} \tag{6}$$

$F_1$-score: The F1-score is the harmonic mean of precision and recall, and it provides a single measure of the overall performance of the model. It is calculated as (7):

$$F_1 = 2 * \frac{Precision * Recall}{Precision + Recall} \tag{7}$$

## 3. Results

### 3.1. CNN Calibration and Verification

The results of the CNN detection are shown in Table 5. The results reported on the training set show that for samples with a label of 0.0, the CNN achieved a precision of 0.94, a recall of 0.84, and an F1-score of 0.89. For samples with a label of 1.0, the CNN achieved a precision of 0.86, a recall of 0.95, and an F1-score of 0.90. The results reported on the test set show that for samples with a label of 0.0, the CNN achieved a precision of 0.88, a recall of 0.86, and an F1-score of 0.84. For samples with a label of 1.0, the CNN achieved a precision of 0.86, a recall of 0.88, and an F1-score of 0.87. Overall, the performance of the model on the training and test sets was comparable, and the F1-scores were high, indicating good performance of the CNN.

**Table 5.** Testing of convolutional neural networks, including precious, recall, and F1-score.

|  | **Precision** | **Recall** | **F1-Score** |
|---|---|---|---|
| Train dataset Report |  |  |  |
| 0.0 | 0.94 | 0.84 | 0.89 |
| 1.0 | 0.86 | 0.95 | 0.90 |
| Test dataset Report |  |  |  |
| 0.0 | 0.88 | 0.86 | 0.84 |
| 1.0 | 0.86 | 0.88 | 0.87 |

### 3.2. Collapse Zoning Map Based on the CNN and IV Method

Based on the trained CNN, collapse susceptibility prediction was carried out on a grid-by-grid basis in the study area, and a susceptibility zoning map was created (Figure 18). The susceptibility values in the study area range from 0 to 1, with higher values indicating higher susceptibility of the grid cell to collapse hazards.

The results (Figures 12 and 18) were reclassified into eight classes and five classes (Figures 19–22), respectively, and were evaluated using ROC curves.

According to Table 6 and Figure 23, the accuracy of the information value method was 85.1% and 85.9%, while the accuracy of the CNN-based approach was 87.9% and 87.4%. The ROC curves based on the CNN performed well for both eight-class and five-class classifications, with larger AUC values compared to those of the information value method. The precision of the CNN-based approach was significantly higher than that of the information value method by 1.5% to 2.8%.

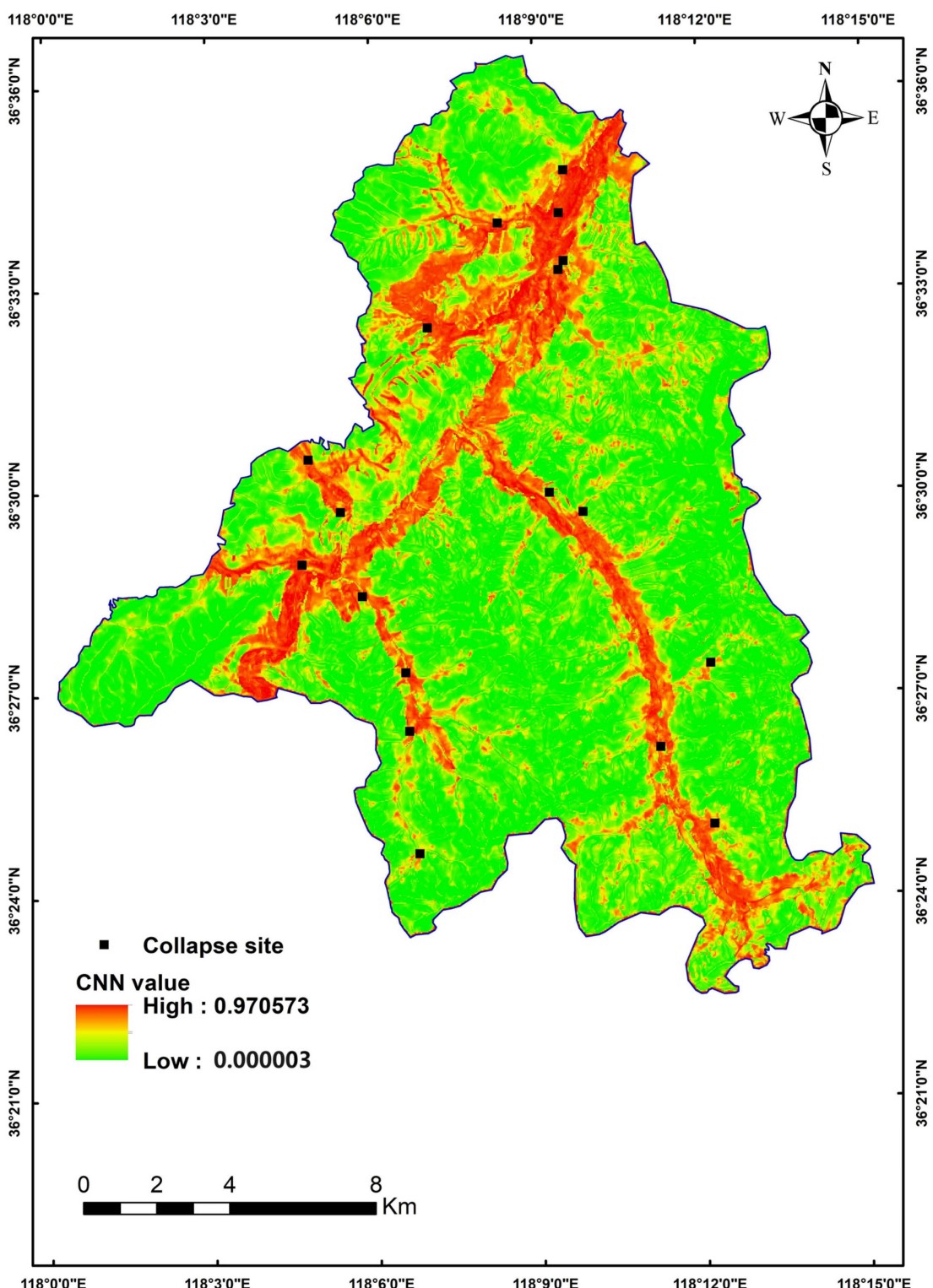

**Figure 18.** Collapse susceptibility zoning map based on the CNN.

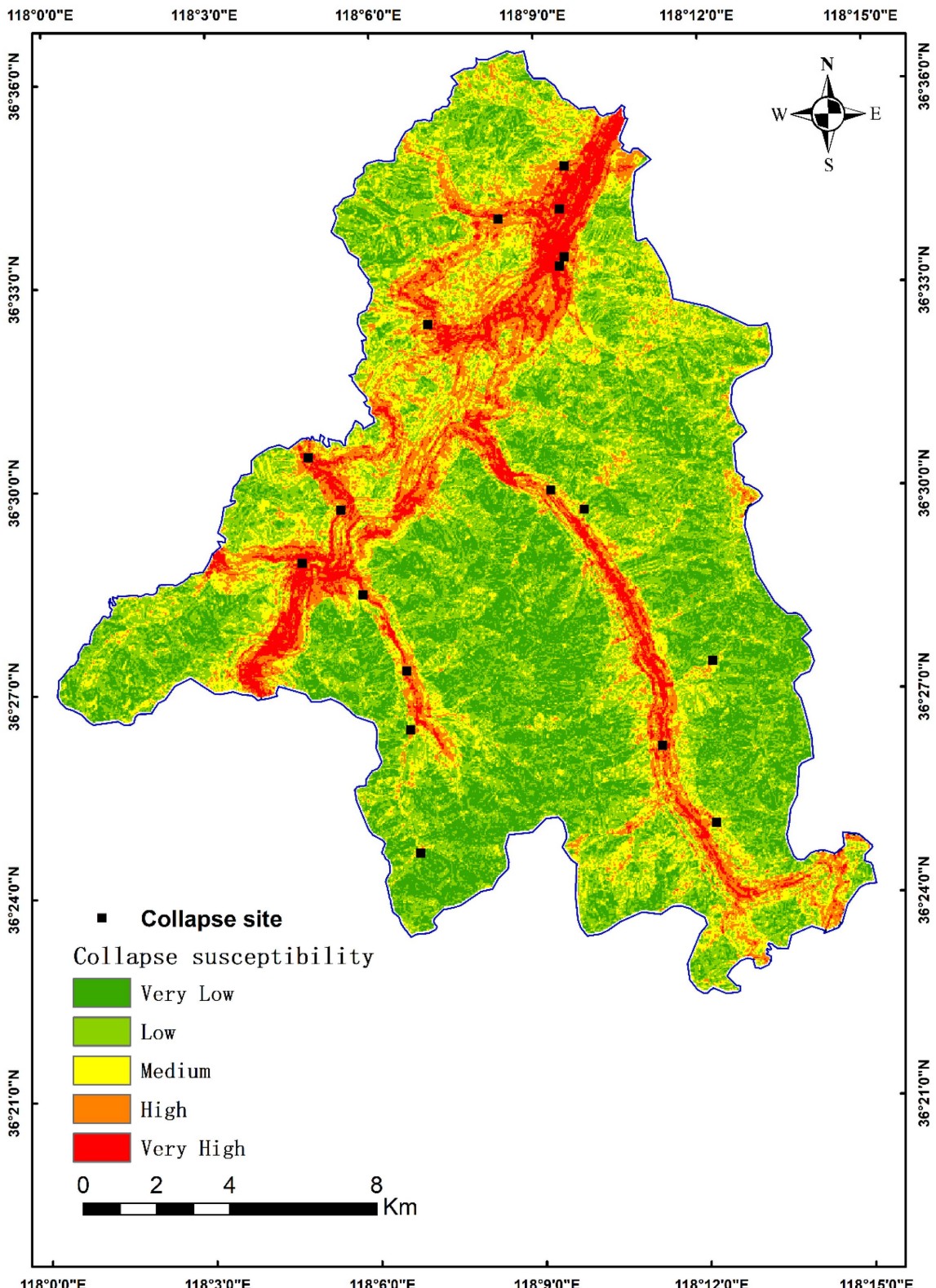

**Figure 19.** The susceptibility maps based on the IV method are divided into five types.

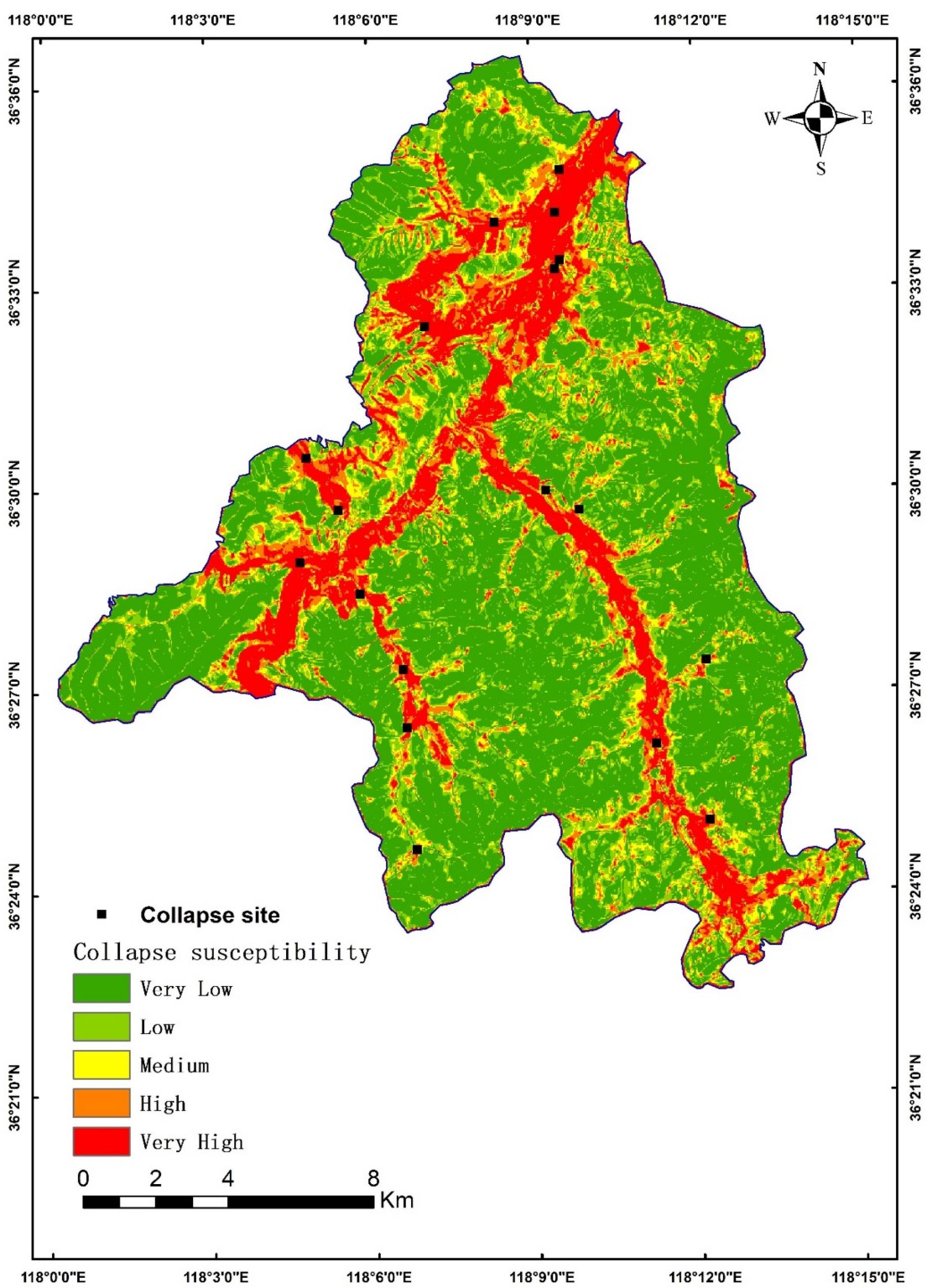

**Figure 20.** The susceptibility maps based on the CNN are divided into five types.

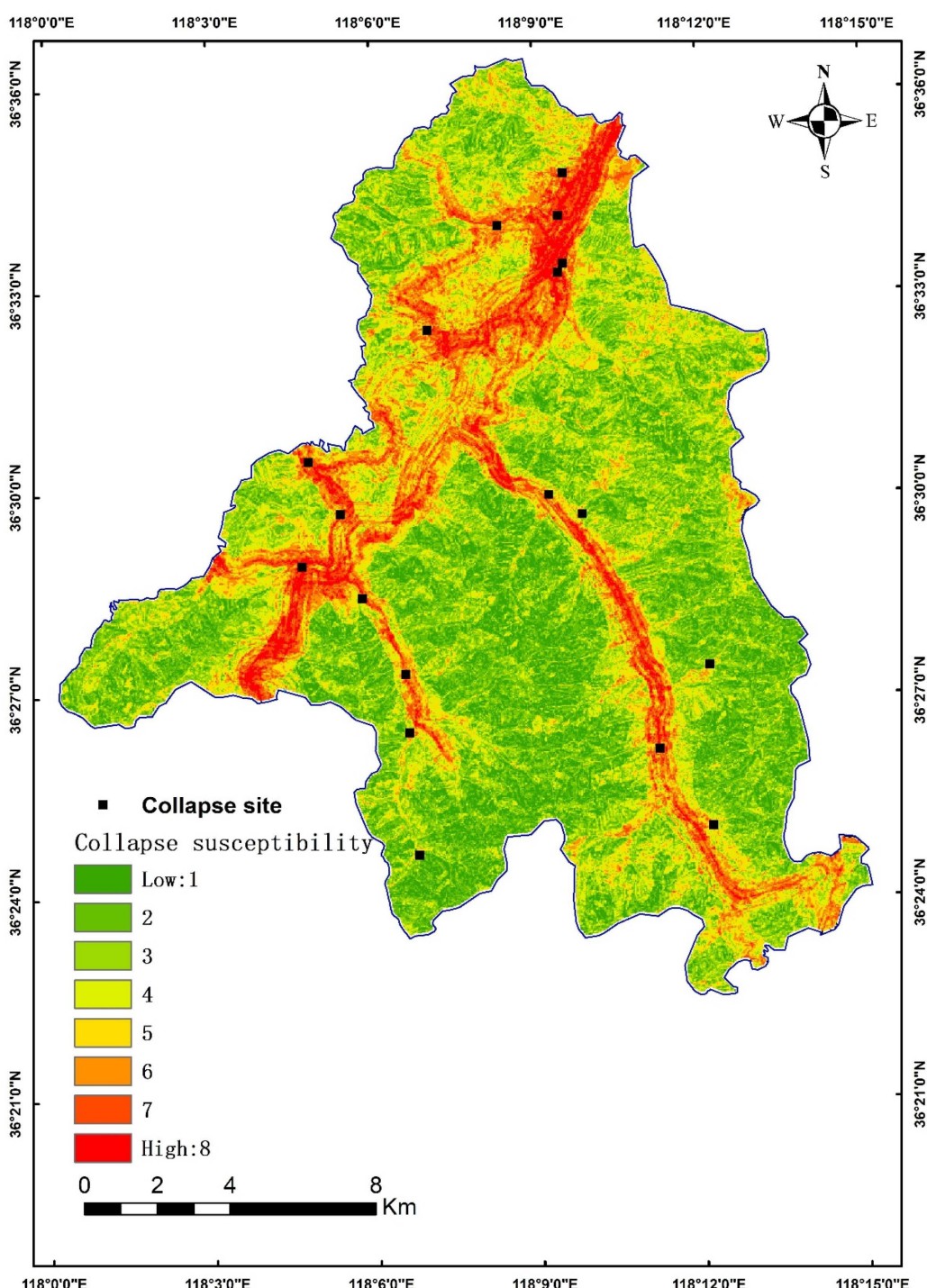

**Figure 21.** The susceptibility maps based on the IV method are divided into eight types.

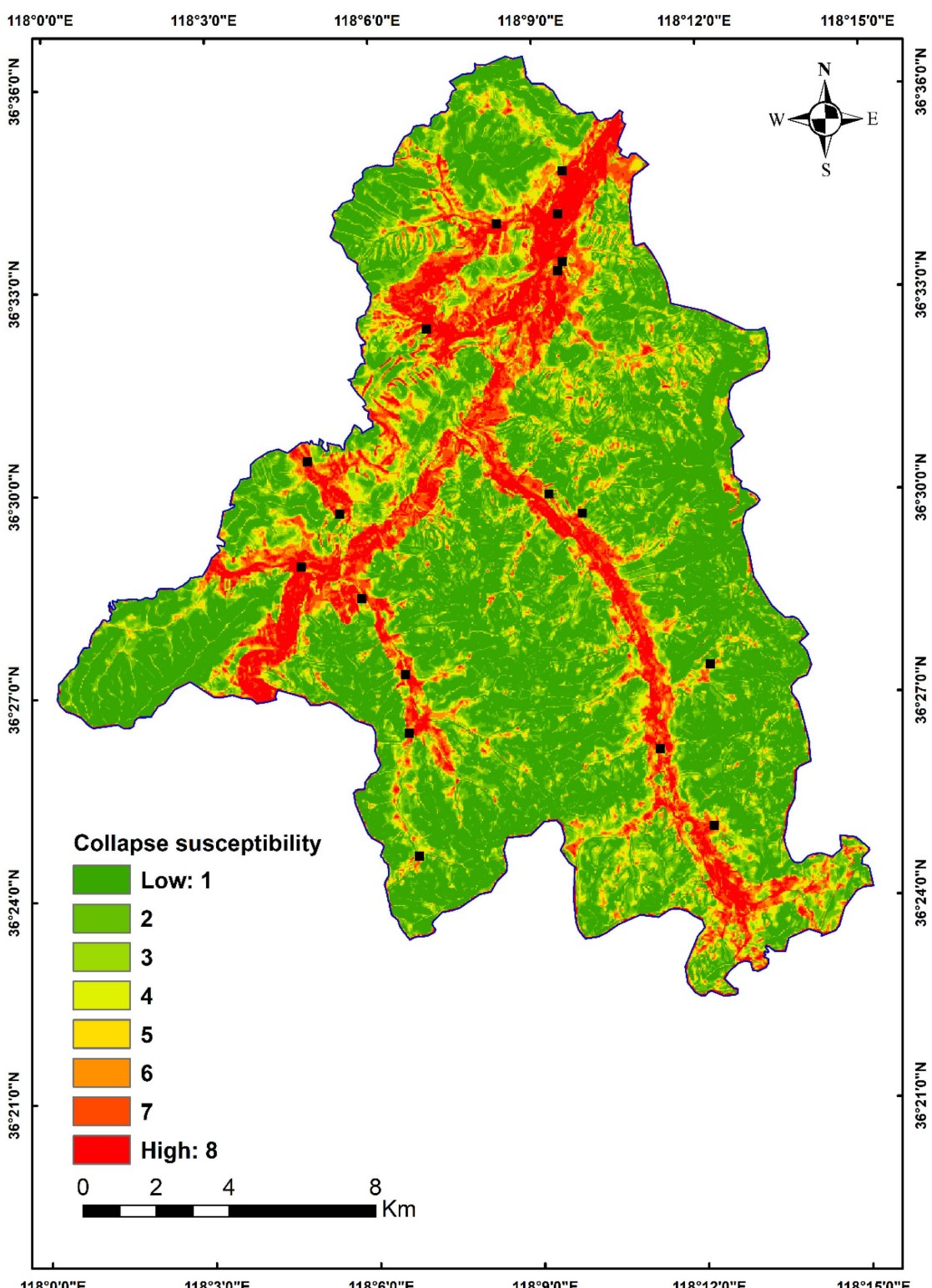

**Figure 22.** The susceptibility maps based on the CNN are divided into eight types.

**Table 6.** Collapse susceptibility assessment frequency ratio.

| Name | Susceptibility | Grid Number | Collapse Number | Grid Ratio | Disaster Proportion | AUC |
|---|---|---|---|---|---|---|
| CNN TO 5 TYPE | 1 | 150,537 | 0 | 0.501 | 0.000 | |
| | 2 | 48,033 | 1 | 0.160 | 0.056 | |
| | 3 | 29,542 | 0 | 0.098 | 0.000 | 0.879 |
| | 4 | 27,408 | 4 | 0.091 | 0.222 | |
| | 5 | 44,958 | 13 | 0.150 | 0.722 | |
| IV TO 5 TYPE | 1 | 73,360 | 1 | 0.249 | 0.056 | |
| | 2 | 100,329 | 1 | 0.341 | 0.056 | |
| | 3 | 63,802 | 2 | 0.217 | 0.111 | 0.851 |
| | 4 | 38,535 | 2 | 0.131 | 0.111 | |
| | 5 | 18,079 | 12 | 0.061 | 0.667 | |
| CNN TO 8 TYPE | 1 | 127,531 | 0 | 0.424427 | 0.000 | |
| | 2 | 45,276 | 1 | 0.15068 | 0.056 | |
| | 3 | 26,382 | 0 | 0.0878 | 0.000 | |
| | 4 | 19,515 | 0 | 0.064947 | 0.000 | |
| | 5 | 16,765 | 1 | 0.055794 | 0.056 | 0.874 |
| | 6 | 16,810 | 1 | 0.055944 | 0.056 | |
| | 7 | 20,138 | 8 | 0.06702 | 0.444 | |
| | 8 | 28,061 | 7 | 0.093388 | 0.389 | |
| IV TO 8 TYPE | 1 | 36,302 | 0 | 0.123432 | 0.000 | |
| | 2 | 63,830 | 1 | 0.217031 | 0.056 | |
| | 3 | 61,603 | 1 | 0.209459 | 0.056 | |
| | 4 | 47,636 | 0 | 0.161969 | 0.000 | |
| | 5 | 34,070 | 2 | 0.115843 | 0.111 | 0.859 |
| | 6 | 24,755 | 2 | 0.084171 | 0.111 | |
| | 7 | 17,358 | 5 | 0.05902 | 0.278 | |
| | 8 | 8551 | 7 | 0.029075 | 0.389 | |

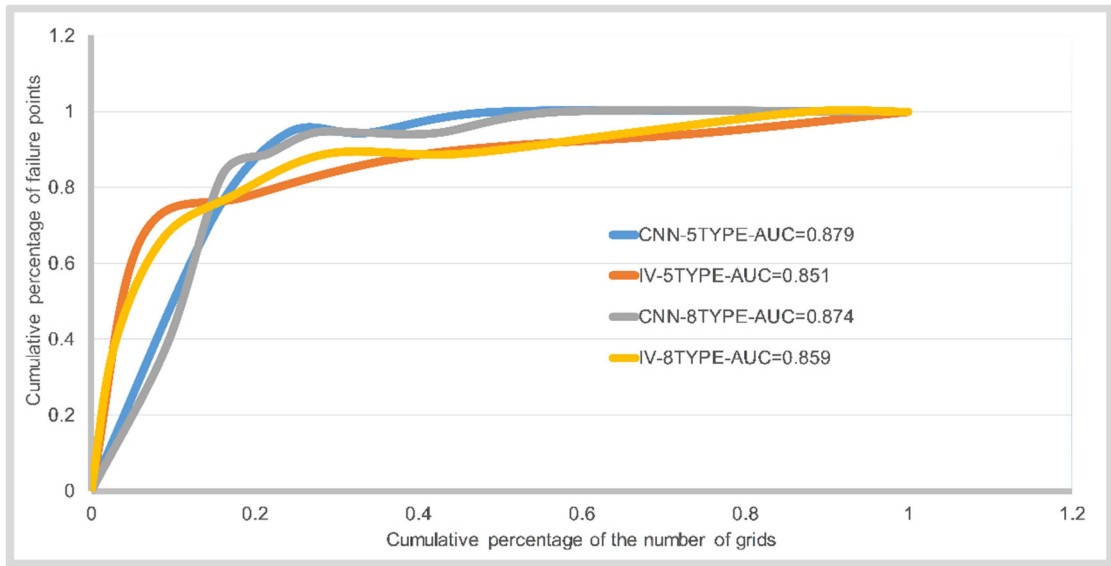

**Figure 23.** ROC and AUC.

## 4. Discussion

The CNN is a deep learning model that is commonly used to analyze visual images, especially in the field of remote sensing images, using two-dimensional convolutional neural networks for landslide and ground object recognition [44–46]. Compared with the traditional method of assigning weight to each influence factor, the CNN has more powerful feature learning functions, such as automatic feature extraction and processing of high-

dimensional data, and can extract abstract information that is difficult to describe directly. Compared with landslides, collapse also has many influencing factors and complicated causes. The CNN has been widely used in landslide susceptibility prediction [47–49], so we try to apply the CNN to the assessment of collapse susceptibility.

Previous studies of geological hazards, especially their susceptibility, have mainly focused on landslides, debris flows, land subsidence, and so on. In this paper, the CNN and IV methods are used to study the collapse hazard of Taihe Town to analyze the applicability of the CNN in the field of collapse. In geohazard susceptibility assessment, the IV method is a common quantitative evaluation method [50,51]. It is based on statistical principles and is used to assess the influence of different factors or variables on geohazard occurrence. The advantages of the IV method lie in its ability to quantitatively assess the contribution of different factors or variables to geohazard occurrence and provide a scientific basis for decision-making. We compare the results based on the information method to prove that the results based on the CNN are equally reliable.

There are many research methods on how to evaluate the vulnerability of geological hazards. How to choose a suitable research method and how to consider the influence of natural conditions and human activities to carry out accurate and reasonable susceptibility regionalization are still major challenges. When utilizing the CNN for collapse susceptibility assessment, accurately selecting the appropriate influencing factors and ensuring the reliability and robustness of the model in the presence of numerous factors are crucial. Here are some approaches to address these issues: ① Prior analysis and feature selection: Conduct a thorough analysis and study of various potential influencing factors before building the CNN model. This includes human activities, geological features, climate conditions, land use, and more. Through in-depth analysis of data and domain knowledge, select the key factors that have the most significant impact during disasters. The feature selection process should be based on scientific principles and expert knowledge to ensure that the chosen influencing factors are relevant and meaningful. ② Integration of multi-source data: Integrate information from multiple data sources, such as remote sensing data, GIS data, human activity data, etc. By leveraging multiple data sources, comprehensive and accurate information about influencing factors can be obtained. This helps improve the reliability and robustness of the model, reducing biases and incompleteness associated with a single data source. ③ Model validation and evaluation: Perform model validation and evaluation to verify its reliability and robustness. Use techniques such as cross-validation, validation sets, and test sets to assess the model's performance on different datasets. This allows for checking whether the model can produce stable and consistent results across multiple datasets and different scenarios. ④ Continuous improvement and updates: Continuous improvement and updates of the model are key to ensuring reliability and robustness as data and domain knowledge accumulate. Regularly review and update the model to adapt to new data, new influencing factors, and new challenges.

The difficulty of neural network construction lies in the adjustment of hyperparameters. How to optimize hyperparameters is very important to improve accuracy [52]. Optimizing hyperparameters refers to adjusting the hyperparameters of a neural network to find the optimal configuration that maximizes the network's performance. Hyperparameters include learning rate, batch size, number of layers, activation functions, and others. They directly affect the training process and performance of the neural network. In this study, based on experience and previous experimental results, select a set of reasonable hyperparameter values. These empirical rules may come from domain knowledge or recommendations from the relevant literature. Although this method is relatively simple, it can yield good results in certain cases.

During the dataset expansion process, we strategically select eight points surrounding each original disaster point. This selection takes into account the proximity to the collapse point as well as the similarity in environmental conditions and characteristics. By doing so, we effectively minimize the introduction of irrelevant information and noise. Hence, we supplement the dataset with these carefully chosen eight points within the grid surrounding

the coordinates of the disaster point. However, the noise generated by these data or the resulting data imbalance still needs to be further quantified.

The incremental method of data points directly affects the accuracy of the results [53]. Moreover, the existing research is challenging to select valuable data. How to choose the points reasonably remains to be studied. An et al. [54] select non-disaster points evenly distributed throughout the region outside the 500 m buffer of the disaster point. Wang et al. [55] found the low-prone areas based on the IV method and randomly selected several non-hazard points in the extremely low and low-prone areas. In this paper, we adopt the most commonly used buffer method, which has universal applicability. This is an oversampling method, the purpose of which is to generate a new sample from an existing few samples. However, the model learning sample noise will reduce the classification accuracy of the model. This paper lacks the study of noise and hopes to describe it in the future.

In this paper, the land cover type data is excluded and NDVI data is retained when the data correlation analysis is carried out. The comparison of land cover types was not made by excluding NDVI data. What kind of data should be retained to improve the accuracy of the results needs further research.

## 5. Conclusions

The CNN has excellent data extraction capabilities and can discover potential functional relationships from complex data. In terms of collapse susceptibility prediction, the CNN can effectively extract advanced features and accurately predict the susceptibility of collapse. In this study, Taihe Town, Zibo City, Shandong Province was selected as the study area, and both the CNN and IV methods were used for collapse susceptibility zoning. In the CNN-based susceptibility assessment, eight influencing factors, including distance to roads and water systems, NDVI, plane curvature, profile curvature, slope, aspect, and geological data, were selected. The raw collapse data was incrementally processed and converted into one-dimensional data, and a CNN structure was constructed for collapse susceptibility analysis. At the same time, the susceptibility map is made by the IV method and compared with the susceptibility map based on the CNN. This paper proves the feasibility of using the CNN to evaluate the collapse susceptibility assessment in the study area. The following conclusions were obtained:

(1) The results of collapse susceptibility assessment based on both the IV and CNN methods can effectively characterize the susceptibility of collapse in the study area, with a large number of collapse points falling in the high susceptibility zones. The accuracy of the CNN-based results was higher than that of the IV method by approximately 1.5% to 2.8%, indicating that the CNN-based results are more accurate and reliable than those obtained using the information value method.

(2) The 1D-CNN structure based on one-dimensional data achieved reliable prediction results in collapse susceptibility assessment, with an accuracy of 87.9% and 87.4%. The one-dimensional data structure can effectively present the relationship between collapse and influencing factors.

(3) This study demonstrated the feasibility of using incremental data in dataset construction (Section 2.3.3). If non-disaster points can be accurately selected or sufficient data is available when expanding the non-disaster points, the accuracy of the results may be further improved.

(4) When the zoning map was reclassified into five or eight classes, the AUC values did not show the same or decreasing trend, indicating that increasing the number of classification data does not necessarily improve the growth rate.

(5) The CNN constructed in this study is not the optimal neural network structure, and if all structures can be exhaustively searched, it may be possible to find model parameters and hyperparameters with higher accuracy.

(6) In this study, the effectiveness of the two methods was compared using ROC curves, and the comparison was not based on the differences in the zoning maps. The next step could be to use new methods to quantitatively characterize the degree of difference between the two susceptibility zoning maps.

In conclusion, both CNN-based and IV-based collapse susceptibility assessments are accurate and reliable. However, there are some challenges when applying the CNN model to collapse susceptibility evaluation, such as the need for sufficient training samples and complex hyperparameter optimization. To address these challenges and explore more effective deep learning models, future research goals should focus on improving the CNN model's performance in collapse susceptibility assessment.

**Author Contributions:** Conceptualization, B.X.H. and B.L.; methodology, S.Z.; software, H.L.; validation, F.M., Y.L. and B.X.H.; formal analysis, H.L.; investigation, Y.L.; resources, B.L.; data curation, S.Z.; writing—original draft preparation, H.L.; writing—review and editing, H.L.; visualization, S.Z.; supervision, Y.L.; project administration, B.L.; funding acquisition, B.L. All authors have read and agreed to the published version of the manuscript.

**Funding:** This research received no external funding.

**Data Availability Statement:** Data is contained within the article.

**Conflicts of Interest:** The authors declare no conflicts of interest.

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
