# Peer review of "Collapse Susceptibility Assessment in Taihe Town Based on Convolutional Neural Network and Information Value Method"

_water, doi:10.3390/w16050709_

Round 1

Reviewer 1 Report

Comments and Suggestions for Authors

The manuscript by Li et al. discusses the complexity of the causal mechanisms behind collapse disasters and highlights the use of Convolutional Neural Network (CNN) for robust feature extraction to simulate and accurately predict collapse threats in Taihe town.

Collapse is a geological disaster. Monitoring, predicting, and implementing timely preventive measures for geological and structural stability are crucial for mitigating the hazards associated with collapses. This research is very important and worthy of publication. I recommend a minor revision.

Comments

1) Line 24: Here the authors may list some typical collapse events to highlight their hazards.

2) Line 36: Here the authors may say that deep learning has been widely applied in disaster warning, for example, volcanic eruptions or tsunamis. The recently published paper on tsunami disaster warning could be referred:

https://doi.org/10.1186/s40623-023-01912-6.

3) Line 92: Where does the data for "field survey results" come from? Please specify explicitly.

4) Line 184: How does geological data provide information in the assessment of collapse susceptibility, particularly regarding the composition and structural characteristics of rocks?

5) Line 195: Regarding the correlation between the "land cover type" dataset and the "Normalized Difference Vegetation Index (NDVI) data," why is it recommended to retain the "NDVI data" and discard the "land cover type" data for improving the accuracy of the evaluation results?

6) Line 208: What is the specific significance of reclassifying, and how can subjective biases be avoided in this process?

7) Line 267: Is the selection of 8 points around each original disaster point during the dataset expansion process likely to introduce excessive noise or result in data imbalance?

8) Line 272: Is the difference between CNN and IV methods related to the size of the CNN training set? If the size of the CNN training set is adjusted, will there be different outcomes? This is crucial as it can help determine the robustness of the method.

9) Line 375: When utilizing CNN for collapse susceptibility assessment, the text mentions significant influence from human activities on disaster points. However, how can one accurately select the appropriate influencing factors, and how can the reliability and robustness of the model be ensured in the presence of numerous influencing factors?

Other Comments

Line 64: Remember to line break before the second point.

Figure 1: "Taihe Town" in the legend should have capital letters for the initial letters.

Comments on the Quality of English Language

Minor editing of English language required.

Author Response

Thank you for offering us an opportunity to improve the quality of our submitted manuscript (water-2803902).  We appreciated the reviewers’ constructive and insightful comments very much.  In this revision, we have addressed all of these comments/suggest-ions.  We hope the revised manuscript has now met the publication standard of your journal.

Reviewer #1

1) Line 24: Here the authors may list some typical collapse events to highlight their hazards.

Response: Thanks for your advice, I added the case on line 28 and marked it in yellow.

2) Line 36: Here the authors may say that deep learning has been widely applied in disaster warning, for example, volcanic eruptions or tsunamis. The recently published paper on tsunami disaster warning could be referred: https://doi.org/10.1186/s40623-023-01912-6.

Response: Thank you for your advice, I have amended it on line 43 and marked it in yellow.

3) Line 92: Where does the data for "field survey results" come from? Please specify explicitly.

Response: Thank you for your advice. We have added data sources on line 103 and marked them in yellow.

4) Line 184: How does geological data provide information in the assessment of collapse susceptibility, particularly regarding the composition and structural characteristics of rocks?

Response:First, Thanks for your advice. We added descriptions and marked them in yellow on line 205. Secondly, let me explain the data we use. Geological researchers determine and assign names to the geological data we utilize based on the principles of stratigraphy and geology. This process involves meticulous observation and analysis of rock types, fossil assemblages, stratigraphic sequences, and stratigraphic characteristics. Furthermore, geologists engage in communication and discussion with their peers to ensure accuracy and consistency in their findings.

5) Line 195: Regarding the correlation between the "land cover type" dataset and the "Normalized Difference Vegetation Index (NDVI) data," why is it recommended to retain the "NDVI data" and discard the "land cover type" data for improving the accuracy of the evaluation results?

Response:Thanks for your suggestion, we have corrected it on line 224 and marked in yellow. We did not compare the effect of retaining what data on the results, which we will explain in the conclusion section.

6) Line 208: What is the specific significance of reclassifying, and how can subjective biases be avoided in this process?

Response: Thanks for your suggestion, We added descriptions and marked them in yellow on line 238.

7) Line 267: Is the selection of 8 points around each original disaster point during the dataset expansion process likely to introduce excessive noise or result in data imbalance?

Response: Thank you for your question. I gave a more accurate description of the selected points on line 305 and marked them in yellow. First, the collapse point of Taihe town is mainly a small collapse, and the specific range of each collapse point is not all in a 30m*30m grid. Secondly, according to the first law of geography, "all things are related to other things, but things near are more related than things far away." By considering points that are close to the collapse point and have similar environmental conditions or characteristics, the introduction of irrelevant information and noise can be effectively reduced. Therefore, we choose the eight selected points within the grid selected around the disaster point coordinates as a supplement. However, the specific amount of noise avoided or removed cannot be described quantitatively, which we will explain in the discussion section.

8) Line 272: Is the difference between CNN and IV methods related to the size of the CNN training set? If the size of the CNN training set is adjusted, will there be different outcomes? This is crucial as it can help determine the robustness of the method.

Response: Thank you for your question. 1.The difference between CNN and IV methods is not directly related to the size of the CNN training set. The size of the training set primarily affects the performance and generalization capability of the CNN model, rather than the fundamental differences between the two methods. 2. However, adjusting the size of the CNN training set can have an impact on the outcomes and robustness of the method. Increasing the size of the training set generally allows the CNN model to learn more representative patterns and generalize better to unseen data. With a larger training set, the CNN model may capture a wider range of features and variations, potentially leading to more accurate and robust predictions. On the other hand, if the size of the CNN training set is small, the model may be more prone to overfitting, where it memorizes the training data instead of learning meaningful patterns. As a result, the CNN model's performance may be limited, and its outcomes may be less reliable and robust when applied to new, unseen data. In summary, while the size of the CNN training set can influence the performance and robustness of the method, it is just one of several factors to consider. Other factors, such as data quality, diversity, and model architecture, also play important roles in determining the overall robustness and effectiveness of the CNN method.

9) Line 375: When utilizing CNN for collapse susceptibility assessment, the text mentions significant influence from human activities on disaster points. However, how can one accurately select the appropriate influencing factors, and how can the reliability and robustness of the model be ensured in the presence of numerous influencing factors?

Response: Thank you for your question. I added the corresponding content in the article on line 431 and marked in yellow.

When utilizing CNN for collapse susceptibility assessment, accurately selecting the appropriate influencing factors and ensuring the reliability and robustness of the model in the presence of numerous factors are crucial challenges. Here are some approaches to address these issues:â‘ Prior analysis and feature selection: Conduct a thorough analysis and study of various potential influencing factors before building the CNN model. This includes human activities, geological features, climate conditions, land use, and more. Through in-depth analysis of data and domain knowledge, select the key factors that have the most significant impact during disasters. The feature selection process should be based on scientific principles and expert knowledge to ensure that the chosen influencing factors are relevant and meaningful. â‘¡ Integration of multi-source data: Integrate information from multiple data sources, such as remote sensing data, GIS data, human activity data, etc. By leveraging multiple data sources, comprehensive and accurate information about influencing factors can be obtained. This helps improve the reliability and robustness of the model, reducing biases and incompleteness associated with a single data source. â‘¢ Model validation and evaluation: Perform model validation and evaluation to verify its reliability and robustness. Use techniques such as cross-validation, validation sets, and test sets to assess the model's performance on different datasets. This allows checking whether the model can produce stable and consistent results across multiple datasets and different scenarios. â‘£Continuous improvement and updates: Continuous improvement and updates of the model are key to ensuring reliability and robustness as data and domain knowledge accumulate. Regularly review and update the model to adapt to new data, new influencing factors, and new challenges.

In conclusion, by employing approaches such as prior analysis, feature selection, integration of multi-source data, model validation and evaluation, and continuous improvement and updates, one can accurately select the appropriate influencing factors and ensure the reliability and robustness of the CNN model. The combination of these approaches will help construct a more accurate, reliable, and adaptive CNN model for collapse susceptibility assessment.

Other Comments

Line 64: Remember to line break before the second point.

Response:Thank you for your suggestion, I have corrected it.

Figure 1: "Taihe Town" in the legend should have capital letters for the initial letters.

Response:Thank you for your suggestion, I have resubmitted the picture.

Reviewer 2 Report

Comments and Suggestions for Authors

1. p.1, line 45, remove one 'disaster'

2. Do authors use landslides and collapse synonymously? Appears so in first paragraph of p. 2

3. p.2, line 93, authors say 53 disaster points. How do authors define disaster and how is it differentiated from hazard

4. Fig. 2 - collapse here shows rockfall - authors must make clear what hazard they are dealing with - landslide, collapse or rockfall?

5. Fig. 3 - the collapse sites are less than the specified 53 points. Why?

6. 2.2.2 - define the water system. is it a stream or river or channel, etc.?

7. Table 1 - correlation matrix -what software was used?

8. 2.3.3 - dataset - is it 18 points, 53 points or 162 points used as training set? How were the 8 points around the disaster site selected?

9. Discussion is not adequate

10. The authors have to clarify this term 'collapse' which is used in different senses throughout the manuscript

11. Authors should emphasize the uniqueness on their work

Comments on the Quality of English Language

The English language is fine. 

Author Response

Thank you for offering us an opportunity to improve the quality of our submitted manuscript (water-2803902). We appreciated the reviewers’ constructive and insightful comments very much. In this revision, we have addressed all these comments/suggest-ions. We hope the revised manuscript has now met the publication standard of your journal.

Reviewer #2

  1. p.1, line 45, remove one 'disaster'

Response: Thanks for your advice, I've removed it

  1. Do authors use landslides and collapse synonymously? Appears so in first paragraph of p. 2

Response: Thank you for your question. I'm not equating a landslide with a collapse. Because the research literature on the use of CNN in collapse is difficult to search, in contrast, there are many studies on the use of CNN for susceptibility in landslide. My example of landslides is that in using CNN in the field of geological hazards, the two are not used equally.

  1. p.2, line 93, authors say 53 disaster points. How do authors define disaster and how is it differentiated from hazard

Response: Thank you for your question. We have fixed the mistake on line 102 and marked in red. Hazards are potential threats or events that can cause harm, while disasters realize those hazards, resulting in significant damage or loss.

  1. Fig. 2 - collapse here shows rockfall - authors must make clear what hazard they are dealing with - landslide, collapse or rockfall?

Response: Thanks to your suggestion, we have added the definition of collapse in line 24. The Figure.2 was resubmitted.

The collapse can be divided into soil collapse and rock collapse in the study area. The specific type of Classification is based on the Standard of Classification for Geological Hazard.

https://www.cgs.gov.cn/ddztt/jqthd/fzjz/xmjz/bzgf/202006/P020200602403973808311.pdf

  1. Fig. 3 - the collapse sites are less than the specified 53 points. Why?

Response: Thank you for your question. We have fixed the mistake on line 102 and marked in red. 53 points belong to Zichuan district’s collapse, landslide and debris flow, is a cumulative value.

  1. 2.2.2 - define the water system. is it a stream or river or channel, etc.?

Response: Thanks for your suggestion, I added the definition of water system on line 141 and marked it in red

  1. Table 1 - correlation matrix -what software was used?

Response: Thank you for your question. We have added the relevant instructions on line 219 and marked in red. This paper uses the Pearson correlation coefficient based on GIS platform to calculate the degree of correlation between sample features.

  1. 2.3.3 - dataset - is it 18 points, 53 points or 162 points used as training set? How were the 8 points around the disaster site selected?

Response: Thank you for your question.

1.There are 324 points in total, including 162 collapse hazard points and 162 non- collapse hazard points. 70% of the data(324points) was used for CNN model training, and 30% of the data was used for CNN model validation.

2.A nine grid, each square represents a 30m*30m grid, the original point (black) in the middle, and then take 8 points on the outside, a total of 162 points (18*8+18=162).

In this picture, the black is the raw data. Red is the expanded data

  1. Discussion is not adequate

Response: Thanks for your suggestions, I have improved the discussion section and marked it in red.

  1. The authors have to clarify this term 'collapse' which is used in different senses throughout the manuscript

Response: Thanks to your suggestion, we have added the definition of collapse in line 24.

  1. Authors should emphasize the uniqueness on their work

Response: Thanks for your suggestion, I have added in the summary and marked it in red

Round 2

Reviewer 2 Report

Comments and Suggestions for Authors

The authors have answered the queries satisfactorily.

Comments on the Quality of English Language

Minor English edits would improve readability of paper